# Elevated 3D structures of PM$_{2.5}$ and impact of complex terrain-forcing circulations on heavy haze pollution over Sichuan Basin, China

Zhuozhi Shu[1,4], Yubao Liu[1,4], Tianliang Zhao[1,4,*], Junrong Xia[1], Chenggang Wang[1], Le Cao[1], Haoliang Wang[1], Lei Zhang[2], Yu Zheng[2], Lijuan Shen[1], Lei Luo[3], Yueqing Li[3]

[1]Collaborative Innovation Center on Forecast and Evaluation of Meteorological Disasters, Key Laboratory for Aerosol-Cloud-Precipitation of China Meteorological Administration, Nanjing University of Information Science and Technology, Nanjing, 210044, China

[2]State Key Laboratory of Disastrous Weather, China Academy of Meteorological Sciences, Beijing, 100081, China

[3]Institute of Plateau Meteorology, China Meteorological Administration, Chengdu, 610072, China

[4]Precision Regional Earth Modeling and Information Center, Nanjing University of Information Science and Technology, Nanjing, 210044, China

*Correspondence*: Tianliang Zhao (tlzhao@nuist.edu.cn)

**Abstract.** Deep basins create a uniquely favorable air pollution causing condition, and the Sichuan Basin (SCB) in Southwest China is such a basin featuring frequent heavy pollution. A wintertime heavy haze pollution event in SCB was studied with conventional and intensive observation data and the WRF-Chem model to explore the three-dimensional distribution of PM$_{2.5}$ to understand the impact of regional pollutant emissions, basin circulations associated with plateaus, and downwind transport to the adjacent areas. It was found that the vertical structure of PM$_{2.5}$ over SCB was characterized by a remarkable hollow sandwiched by high PM$_{2.5}$ layers at heights of 1.5–3 km and a highly polluted near-surface layer. The southwesterlies over the Tibetan Plateau (TP) and Yunan-Guizhou Plateau (YGP) resulted in a lee vortex over the SCB, which helped form and maintain heavy PM$_{2.5}$ pollution. The basin PM$_{2.5}$ was lifted into the free troposphere and transported outside of the SCB. At the bottom of the SCB, high PM$_{2.5}$ concentrations were mostly located in the northwestern and southern regions. Due to the blocking effect of the plateau terrain on the northeasterly winds, PM$_{2.5}$ gradually increased from northeast to southwest in the basin. In the lower free troposphere, the high PM$_{2.5}$ centers were distributed over the northwestern and southwestern SCB areas, as well as the central SCB region. For this event, the regional emissions from SCB contributed 75.4–94.6 % to the surface PM$_{2.5}$ concentrations in SCB. The SCB emissions were the major source of PM$_{2.5}$ over the eastern regions of

the TP and the northern regions of YGP, with contribution rates of 72.7 % and 70.5 %, respectively, during the dissipation stage of heavy air pollution over SCB, which was regarded as the major pollutant source affecting atmospheric environment changes in Southwest China.

## 1 Introduction

Haze pollution has caused serious environmental problems, especially in the densely populated and economically developed regions in China, which have high levels of fine particulate matter ($PM_{2.5}$) (particulate matter with an aerodynamic diameter equal to or less than 2.5 μm) (Guo et al., 2014; Li et al., 2015; Gu and Yim, 2016; Lin et al., 2018). Owing to the significant adverse effects on human health and climate change (Dawson et al., 2007; Langrish et al., 2012; Megaritis et al., 2014; Guo et al., 2016), understanding $PM_{2.5}$ pollution distributions and mechanisms is of high interest in environmental and climate studies.

Anthropogenic pollutant emissions and stagnant meteorological conditions are commonly regarded as two key factors influencing haze pollution with excessive concentrations of $PM_{2.5}$ (Yim et al., 2014; Zhang et al., 2015; Cai et al., 2017). With strong anthropogenic emissions and favorable meteorological conditions, four main regions with frequent heavy haze pollution have been identified, centered over the North China Plain (NCP) (Tao et al., 2012; Ye et al., 2016; Zhang et al., 2016; Huang et al., 2017), the Yangtze River Delta (YRD) in East China (Wang et al., 2012; Li et al., 2015; Tang et al., 2015; Ming et al., 2017), the Pearl River Delta (PRD) in South China (Wu et al., 2013; Zhang et al., 2013; Zhang et al., 2014; Guo et al., 2016), and the Sichuan Basin (SCB) in Southwest China (Tao et al., 2013; Chen and Xie, 2014; Zhou et al., 2019). Haze pollution over the NCP, YRD, and PRD, the main economic centers with large flatlands, has been extensively studied. However, air pollution in the SCB region with highly frequent heavy $PM_{2.5}$ pollution has not been completely understood owing to the complex deep basin terrain, particularly the effect of the immediately adjoining Tibetan Plateau (TP).

The TP's "harbor effect" on the tropospheric westerlies favors a stable atmospheric stratification and low wind speeds in the boundary layer over the downstream SCB (Xu et al., 2015; Xu et al., 2016), which is conducive to air pollutant accumulation in SCB (Yim et al., 2014; Xu et al., 2016; Wang et al., 2018). The downslope flows at the lee side of the plateau can induce a special stagnation

meteorological condition in the lower troposphere (Wang et al., 2015; Ning et al., 2018a). Air stagnation days account for 76.6 % of the total days in winter over the SCB (Liao et al., 2018), where near-surface weakened wind, strong vertical air temperature inversion, and shallow boundary layer significantly restrain the atmospheric diffusion capacity (Ning et al., 2018a; Wang et al., 2018; Tian et al., 2019), resulting in the occurrence of heavy air pollution in the SCB.


The SCB, covering 260,000 $km^2$ of the Sichuan-Chongqing plain with a dense population of more than 100 million people, is a deep basin in Southwest China surrounded by plateaus and mountains. It lies immediately to the east of the TP, with a large elevation drop exceeding 3000 m over a short horizontal distance. The unique terrain effect generates the asymmetries of meteorological and air pollutant distribution (Zhang et al., 2019), with a remarkable difference in $PM_{2.5}$ concentrations between the eastern and western regions over the SCB (Chen and Xie, 2012; Ning et al., 2018b). The weak vertical diffusion in the atmospheric boundary layer is one of the main causes of air pollution in winter (Ye et al., 2013; Hu et al., 2014; Tian et al., 2017; Zhao et al., 2018). Many studies have suggested that air pollution over SCB is mostly caused by the accumulation of air pollutants originating from local emissions (Chen et al., 2014; Liao et al., 2017; Wang et al., 2018; Qiao et al., 2019). However, because of the complex flows in SCB, it is important to study how $PM_{2.5}$ is circulated three-dimensionally to estimate the roles of local emissions and exchanges with outside regions more accurately.



In this study, observation data analysis and numerical experiments were conducted to analyze the three-dimensional distribution of $PM_{2.5}$ concentrations in SCB during a heavy haze pollution episode in January 2017. The contributions of the SCB pollutant emissions and $PM_{2.5}$ transport to the surrounding plateaus and mountains were estimated. Section 2 introduces the observation data and the modeling methods used in this study. Section 3 characterizes the horizontal and vertical distributions of $PM_{2.5}$, during the formation, maintenance and dissipation stages of the heavy haze pollution episode. We also assessed the contribution of local emissions to the heavy $PM_{2.5}$ pollution within SCB, and the impact of external transport of the $PM_{2.5}$ in SCB on the surrounding areas in Southwest China. The summary and conclusions are provided in Section 4.



## 2. Data and model

### 2.1 Observation data

The surface air pollutant concentrations and meteorological elements observed in 18 cities (Fig. 1; Table 1) over SCB were used to investigate the distribution of $PM_{2.5}$, weather circulations, and modeling performance. The hourly meteorological observational data, including surface air temperature, relative humidity, wind speed, and wind direction, were obtained from the Chinese meteorological monitoring network, and the hourly observational $PM_{2.5}$ concentrations were obtained from the China

National Environmental Monitoring Center (http://www.cnemc.cn).

      In addition to the above-mentioned conventional observations, sounding observations were conducted every 3 h using a kite balloon with the sounding system TT12 DigiCORA (Vaisala, Finland), at the Meteorological Observatory of Chengdu (Site 1 in Fig. 1) during 1−20 January 2017. The vertical sounding data of air temperature, wind speed, wind direction, and relative humidity were

observed at time intervals of 1 s. In addition, a micro pulse lidar type 4 system (MPL-4B-IDS, Sigma Space, America) was operated at the observational site, Ya'an (Site 15 in Fig. 1), in the western SCB edge to retrieve the vertical $PM_{2.5}$ structures at 532 nm (laser emission wavelength), 2500 Hz (laser repetition rate), and 6−8 μJ (optimal laser output range).

### 2.3 Model configuration and simulation experiments

The Weather Research and Forecasting with Chemistry (WRF-Chem, version 3.8.1) model was employed to simulate severe haze pollution events over 2–7 January 2017 in SCB (Fig. 2). The spin-up time of modeling for the first 24 h, starting on 31 December 2016, was dropped. The ERA-Interim meteorological reanalysis data of the European Center for Medium-Range Weather Forecasts (https://www.ecmwf.int/en/forecasts/datasets/reanalysis-datasets/) served as the initial and boundary

conditions of the WRF-Chem simulation. The model domains and topography are presented in Fig. 1. There were three nesting domains with domain 1 (D1: 101 × 95) covering most areas of China, domain 2 (D2: 169 × 169) covering Southwest China, and an inner domain 3 (D3: 229 × 213) covering the

SCB and the surrounding areas, at grid intervals of 48, 12, and 3 km, respectively (Fig. 1). Considering the sharp drop of topography between the SCB and surrounding plateaus and mountains with the

terrain influence on westerlies and atmospheric circulations, we adopted the 1:4 grid ratio for simulation experiments. Although an even nested ratio, which is not recommended, may result in interpolation errors at the nested-domain boundary owing to the nature of Arakawa C-grid staggering, the 1:4 grid ratio available in the WRF framework was reasonably applied in previous simulation studies (Tie et al., 2010; Rai et al., 2019). An adequate vertical resolution is

fundamental for evaluating thermal stratification over complex terrain. Therefore, 35 vertical layers from the ground to the model top at 100 hPa were set for the modeling experiments in this study on air pollution change with the 18 layers in the fine resolutions of 30–120 m vertically from the ground to 1 km within the atmospheric boundary layer. The MYJ is a local closure scheme (Janjić, 1994), which is applicable to the atmospheric environment with stable stratification for

weak turbulent mixing (Jia and Zhang, 2020) and underlying complex terrain (Lu et al., 2012; Madala et al., 2015; Bei et al., 2019). Therefore, The MYJ was used as the planetary boundary layer parameterization scheme in the simulation. The detailed physical and chemistry schemes for the WRF-Chem simulation are listed in Table 2, involving the Morrison 2 microphysics (Morrison et al., 2009), the RRTM longwave radiation (Mlawer et al., 2000), the RRTMG shortwave

radiation (Iacono et al., 2008), Noah land surface model (Tewari et al., 2004) and the Grell 3D cumulus (Grell and Devenyi, 2002).

The regional acid deposition model, version 2 (RADM2) (Stockwell et al., 1990) was selected for the atmospheric gas-phase chemistry mechanism, including the main inorganic ions, elemental carbon, primary and secondary organic aerosols, and other aerosol species (Tuccella et al., 2012). The

Multi-resolution Emission Inventory for China (MEIC) from 2012 (http://www.meicmodel.org) with a horizontal resolution of $0.25 \times 0.25°$ was used to model the anthropogenic emissions of air pollutants. The Model of Emissions of Gases and Aerosols from Nature (v2.1) was applied to the natural emission sources in the simulation with dust emission parameterization.

The weakened vertical diffusion capacity is conducive to the accumulation of air pollutants (Ren

et al., 2019) and in the formation of severe haze pollution with the explosive growth of $PM_{2.5}$ (Zhong et al., 2018). Approximately 80 % reduction in turbulent diffusion coefficient provides a more accurate

prediction of $PM_{2.5}$ over the NCP (Wang et al., 2018). High $PM_{2.5}$ levels in the atmosphere could significantly reduce the near-ground solar radiation, resulting in decreasing vertical turbulent diffusion in the boundary layer (Wang et al., 2019), which could be incompletely considered in the

meteorological reanalysis data driving the WRF-Chem simulation. In this study, we found that a 50 % decrease of the turbulent diffusion coefficient in the SCB could greatly improve the deviation of $PM_{2.5}$ simulations in the extremely stable atmosphere through the sensitivity tests. Hence, the turbulent diffusion coefficient was cut halfway for the simulation of the 3D structures of $PM_{2.5}$, during the heavy air pollution event over the SCB region. Two simulation experiments were conducted: 1) a baseline

simulation (Emi-Real), with the MEIC anthropogenic emission inventory over all three domains, and 2) a sensitivity simulation (Emi-Non), similar to Emi-Real but involved shutting down the anthropogenic emission sources in the SCB (Fig. 1). By comparing the $PM_{2.5}$ concentrations between Emi-Real and Emi-Non, we quantified the contribution of local emission sources to the heavy haze pollution over the SCB and estimated the transport from the polluted SCB to the adjoined areas over the eastern TP, the

northern YGP, and the Daba Mountain (DBM) region (Fig. 1). The definite ranges of the three regions were defined with the altitudes of 750–3500 m over 30.5–33.0° N, 102.7–105.3° E (the eastern TP edge), 750–3000 m over 27.8–29° N, 103.5–108.5° E (northern YGP edge), and above 750 m over 31.5–33.0° N, 106.0–109.4° E (DBM region), as shown in Fig. S5.

**2.4 Case description**

A severe haze pollution event occurred during 2−7 January 2017 in the SCB. As shown in Fig. 2, high and low $PM_{2.5}$ concentrations were centered in the western and eastern regions during the episode, respectively, presenting a generally asymmetric horizontal distribution.

Based on the National Ambient Air Quality Standards of China by the Ministry of Ecology and Environment in 2012 (http://www.mee.gov.cn/), light and heavy air pollution levels of $PM_{2.5}$ were

categorized with daily average $PM_{2.5}$ concentrations exceeding 75 and 150 μg m$^{-3}$ in ambient air, respectively. The most heavily polluted regions were mainly concentrated in the northwestern city cluster of SCB, including Chengdu, Deyang, and Meishan, with daily mean $PM_{2.5}$ concentrations exceeding 150 μg m$^{-3}$ (Figs. 1 and 2a). An hourly $PM_{2.5}$ peak of 345.0 μg m$^{-3}$ was observed in Chengdu, a representative megacity in Southwest China. According to the hourly $PM_{2.5}$ variations in the city

cluster over the northwestern SCB region, we divided the heavy haze episode into three periods, P1, P2, and P3, corresponding to formation (from 12:00 p.m. on 2 January to 0:00 on 5 January 2017), maintenance (from 0:00 a.m. on 5 January to 12:00 on 6 January 2017), and dissipation (from 12:00 p.m. on 6 January 0:00 a.m. on 6 January 2017) stages, respectively (local time was used in this study). As shown in Fig. 2b, during P1, the surface $PM_{2.5}$ concentrations sharply increased to the heavy haze pollution level, and then fluctuated at the heavy pollution level in P2. Finally, in P3, the concentrations of $PM_{2.5}$ dropped below 75 μg m$^{-3}$ and the event ended on 7 January 2017 (Fig. 2b).

Under the typical Asian monsoon climate in January over the SCB, the change of synoptic conditions during the haze event over the SCB was characterized by the cold air invasion driven by near-surface northeasterly winds with the vertical configuration of trough development and movement in the mid-latitude westerlies at 700 hPa (Figs. 3 and 7). A 700 hPa trough in the mid-latitude westerlies moved eastward from the eastern edge of the TP to the western SCB margin during P1 (Fig. 3a), the trough of low pressure evolved at 700 hPa during P2 (Fig. 3b), and the 700-hPa trough and the low-pressure system disappeared in P3 over the SCB (Fig. 3c). Meteorologically, the direction and intensity of cold air invasion with near-surface northeasterly winds are steered by the development and movement of the westerly trough in the mid-troposphere (Fig. 3c). The 700-hPa trough approached, developed and disappeared in P1, P2 and P3 of the haze pollution event over the SCB (Fig. 3), which is associated with the increase of northeasterly winds for the cold air invasion to the SCB region during the dissipation stage. The changes in atmospheric circulations in the three stages reflected the meteorological modulation of heavy haze development over the SCB in association with the effect of TP topography on the westerlies.

Analysis of the observations revealed noteworthy patterns of the spatial distribution of surface $PM_{2.5}$ concentrations over SCB in the three periods (Fig. 4). During P1, the surface $PM_{2.5}$ concentrations were distributed relatively even over SCB, but during P2, the $PM_{2.5}$ concentrations exhibited a northeast-southwest gradient and a dramatic increase in the western SCB area. For example, the surface $PM_{2.5}$ concentrations increased from 202.1 to 276.6 μg m$^{-3}$, from 148.6 to 181.0 μg m$^{-3}$, from 104.9 to 205.7 μg m$^{-3}$, and from 145.6 to 168.4 μg m$^{-3}$ at sites 1, 3, 6, and 15, respectively (Fig. 1; Table 1). In contrast, during the dissipation period P3, strong northeasterly winds developed, and the air quality was improved from the northeast to the southwest regions, with the reduction in $PM_{2.5}$

concentrations in the northeastern SCB (Fig. 4). The northeast-southwest gradients of the surface PM$_{2.5}$

concentrations in the SCB mostly resulted from the near-surface northeasterly winds that were blocked

by plateaus and mountains located to the southwest of the SCB, which will be further discussed in the

following sections.

## 3. Results and discussion

### 3.1 Model evaluation

First, we validated the WRF-Chem simulation performance by comparing with the meteorological

and PM$_{2.5}$ observations in the SCB, especially with the intensive vertical soundings, for verifying the

vertical structures of the simulated boundary layer (Fig. S1-S3). The simulated vertical PM$_{2.5}$

distribution in the lower troposphere was evaluated using ground-based MPL detection at site 15 in the

western SCB (Fig. 1, Table 1).

A reasonable simulation of meteorology is crucial for modeling variations in air pollutants (Hanna

et al., 2001). The meteorological simulation was validated by comparing the model results with hourly

surface meteorological observations of 2 m air temperature (T2), 10 m wind speed (WS10), and

relative humidity (RH). The statistical metrics of comparisons between simulated and observed

meteorological variables are given in Table 3, including the mean bias (MB), mean error (ME), and

root mean squared error (RMSE). The verification metrics in Table 3 showed a reasonably good model

performance with reference to previous studies (Emery et al., 2001; Chang and Hanna, 2004), although

RH was slightly underestimated and wind speed was slightly overestimated. The statistical verification

of the simulated surface PM$_{2.5}$ concentrations are shown in Table 4 with the normalized mean bias

(NMB), normalized mean error (NME), mean fractional bias (MFB), and mean fractional error (MFE)

in two levels of light PM$_{2.5}$ pollution (75−150 μg m$^{-3}$) and heavy PM$_{2.5}$ pollution (> 150 μg m$^{-3}$). In

general, the verification suggested that the WRF-Chem simulations reasonably reproduced the

meteorological conditions and the evolution of PM$_{2.5}$ concentrations over SCB, within the criteria for

regulatory applications (Emery et al., 2017).

The vertical structure of the atmospheric boundary layer directly affects the vertical diffusion of

atmospheric pollutants. Therefore, we compared the vertical profiles of the model simulation with the

intensive sounding observations in terms of the variation range and average profiles during the heavy haze episode. Compared with the observed air temperature, the WRF-Chem simulations were evaluated to reasonably capture the vertical temperature profiles for understanding atmospheric stability in the vertical thermodynamic structures of the boundary layer over the SCB (Fig. S3). The potential temperature, wind speed, and RH of the simulation were also validated for both daytime and nighttime, as shown in Fig. 5. The simulated vertical profiles of the meteorological variables were generally acceptable in the lower troposphere (Fig. 5). It should be noted that the significant underestimation of RH above 1 km, where the observed RH reached nearly 100%, was caused by the clouds due to the abundant moisture at night, which the model failed to reproduce.

The MPL-4B lidar, located at site 15 (Fig. 1) on the western edge of the SCB to the east of the TP, continuously detected aerosol extinction ratios in the troposphere. The vertical distribution of the $PM_{2.5}$ mass concentrations was derived from the extinction ratio (Ansmann et al., 2012; Córdoba-Jabonero et al., 2016). The height-time cross-section of the derived and simulated $PM_{2.5}$ mass concentrations from 7:00 a.m. to 2:00 p.m. on 5 January 2017, are presented in Fig. 6. It can be seen that a good agreement

between the lidar observation and the WRF-Chem simulation was achieved. One of the significant features is that in addition to the occurrence of near-surface high $PM_{2.5}$, which is typical for most heavy haze pollution events over areas with a relatively flat terrain, a layer of high $PM_{2.5}$, developed between 1 and 2 km above ground level (Fig. 6a), leaving a hollow layer between the two heavily polluted layers. The upper high $PM_{2.5}$ layer was built due to the uplifting and then overturning of the air flows

associated with the blocking effect of the TP terrain, which is addressed in the next section.

**3.2 Surface $PM_{2.5}$ concentrations**

Figure 7 shows the simulated surface $PM_{2.5}$ concentrations and near-surface wind fields during the formation, maintenance, and dissipation periods of 2−7 January 2017. The high $PM_{2.5}$ concentrations were mostly centered in the northwest and southern SCB regions, featuring the Chengdu-Chongqing

urban agglomeration (Fig. 1). The prevailing northeasterly winds strengthened gradually over the SCB from the P1 to the P2 and P3 periods (Fig. 7). The high plateaus and mountains, especially YGP and TP to the west of the SCB, blocked the upcoming northeasterly winds. The spatial distribution of surface $PM_{2.5}$ concentrations (Fig. 7) clearly reflects the combined effect of the urban anthropogenic air

pollutant emissions and the PM$_{2.5}$ accumulation by the flow convergence forced by the TP and the YGP

blocking the prevailing winds. During the formation and maintenance stage, the surface winds were

weak (1.4–1.7 m s$^{-1}$) over the SCB, and were insufficient to dispel the air pollutants, but led to an

accumulation of PM$_{2.5}$, locally from light to heavy pollution conditions (Fig. 7a, Fig. 7b). During P2,

heavy air pollution blanketed a large area in SCB with excessive PM$_{2.5}$ concentrations (mostly > 150.0

μg m$^{-3}$). During P3, the northeasterly winds intensified and removed PM$_{2.5}$ from the SCB (Fig. 7c).

**3.3 Vertical structures of PM$_{2.5}$ concentrations**

The high terrain of the YGP and TP blocked the northeastern airflows over the SCB by lifting the

airflow along with air pollutants, altering the vertical PM$_{2.5}$ distribution. Therefore, it was of great

interest to analyze the vertical distribution and transport structures of PM$_{2.5}$ over the SCB and the

surrounding regions.

The terrain effect of TP, the "world roof" on the mid-latitude westerlies could modulate haze

pollution in the downstream region over China (Xu et al., 2016). The SCB is immediate to the east of

the TP, with a large elevation drop exceeding 3000 m over a short horizontal distance. The unique

terrain-forcing circulations generate asymmetries in meteorological and air pollutant distributions over

the SCB (Zhang et al., 2019). Chengdu (site 1: 104.02° E; 30.67° N) which is a metropolis in SCB with

high anthropogenic pollutant emissions and has the highest pollution levels in Southwest China (Ning

et al., 2018b), situated on the far west side of the SCB, was selected to better understand the elevated

3D structures of PM$_{2.5}$. It is important to investigate how the urban surface high PM$_{2.5}$ levels evolved

vertically in the atmosphere with the combination of high urban emissions and TP's terrain-forcing

lifting over SCB.

The distributions of PM$_{2.5}$ and the atmospheric circulations in the vertical-meridional and

vertical-zonal cross-sections over the SCB and surrounding areas, respectively, are shown in Figs. 8

and 9 for a clean environment, formation, maintenance, and dissipation periods of the heavy haze

pollution episode. A remarkable feature in the vertical distributions of PM$_{2.5}$ was the unique hollows

over the SCB region, between the two high PM$_{2.5}$ layers at the surface and heights of 1.5–3 km.

Similarly, PM$_{2.5}$ elevated to the free atmosphere in a clean environment and dissipation periods and

pressed down in formation and maintenance periods. The special phenomenon was developed by the

interaction of atmospheric circulations in the free troposphere and topographic forcing in the boundary layer. In the atmospheric boundary layer, the leeside vortices often occur over the SCB owing to the effect of the large TP topography on the mid-latitude westerlies, which reinforces the vertical exchange of $PM_{2.5}$ concentrations (Zhang et al., 2019). Meanwhile, a strong temperature inversion appeared and acted as a lid covering $PM_{2.5}$ due to the trough of a low-pressure system (Ning et al., 2018a). In the current case, the variations of lee vortex circulation, working together with the basin near-surface flows, drove a 3D $PM_{2.5}$ transport and its temporal changes over the SCB (Figs. 8–9).

Driven by the near-surface northeasterly winds (Fig. 7), the high concentrations of near-ground $PM_{2.5}$ over the SCB were uplifted over the windward slopes of TP and YGP, respectively. Comparing the vertical structures of $PM_{2.5}$ and the circulations in different periods, the southwesterly wind prevailed at approximately 3 km height in the clean and dissipation stages (Figs. 8a, 8d, 9a, and 9d). It means that the elevated $PM_{2.5}$ transport process in the free troposphere is a general pattern with the plateau-basin configuration over the SCB. By contrast, the so-called lid with a southwesterly wind in vortex circulation was pressed down to 2 km over the SCB. The uphill airflows were restrained and overturned below 2.5 km (a.s.l.), forming a well-structured vertical sub-circulation over the SCB region (Figs. 8b, 8c, 9b and 9c). Governed by the vertical sub-circulations, the downward transport from the high $PM_{2.5}$ layers could replenish the surface $PM_{2.5}$ concentrations in the northwest SCB with the addition of near-surface accumulation and maintenance of air pollutants. The sink momentum of vertical sub-circulation was weakened and confined $PM_{2.5}$ exchange along with the eastern TP while the trough evolves into a low-pressure system (Figs. 3b, 8c and 9c). The cold air invasion with stronger near-surface northeasterly winds steered by the movement of a westerly trough in the mid-troposphere during the dissipation stage of $PM_{2.5}$ pollution (Figs. 3c, 8d and 9d). The haze pollution event over SCB integrally indicated the formation of elevated $PM_{2.5}$ structure and reconstruction of trans-boundary transport pattern of $PM_{2.5}$.

The TP and YGP lee vortices over the SCB also modify the vertical thermo-dynamical structures in the atmosphere (Xu et. al., 2016), altering the height and intensity of the lid in stable stratification and covering air pollutants (Ning et al., 2018a). The potential temperature vertical gradients (Fig. 5), which are used for assessing atmospheric stability, were estimated with 4.0 K/km, 7.8 K/km, and 5.2 K/km in the boundary layer during the three periods of haze pollution with near-surface strong

temperature inversion (Fig. S3), presenting a thermodynamic structure with stable stratification in the atmospheric boundary layer, weakening the air pollutant dispersion.

The complex terrain-forcing circulations along the windward slopes of TP and YGP, accompanying the lowering of westerlies lid, drove a remarkable hollow of $PM_{2.5}$ sandwiched by a high

$PM_{2.5}$ layer in the free troposphere and a highly polluted near-surface layer in northwest SCB. It was also noted that the high $PM_{2.5}$ concentrations over the SCB were transported to the downwind regions following westerlies by a special pathway above the atmospheric boundary layer.

### 3.4 Distribution of $PM_{2.5}$ in upper high concentration layer

This section describes the characteristics of the upper-layer high $PM_{2.5}$ concentrations. The $PM_{2.5}$

concentrations were averaged between the heights of 1.5–2.5 km, as shown in Fig. 10. Compared with the surface $PM_{2.5}$ concentrations, the $PM_{2.5}$ concentrations decreased significantly in the lower free troposphere (Figs. 7 and 10), reflecting the important role of surface air pollutant emissions in the atmospheric environment over SCB. During the formation period of heavy air pollution event, the $PM_{2.5}$ particles in the free troposphere were concentrated in the northwestern SCB (Fig. 10a). During

P2, high $PM_{2.5}$ centers were developed in the northwestern SCB edge, and $PM_{2.5}$ concentrations increased in the southwestern and central SCB regions (Fig. 10b), reflecting the strong vertical diffusion of $PM_{2.5}$ in the lower troposphere during the heavy air pollution periods (Figs. 8c and 9c). Driven by strong northeasterly winds during P3 (Fig. 7c), the high $PM_{2.5}$ concentrations in the lower free troposphere were centered in the narrow southwestern and southern SCB areas (Fig. 10c), where

$PM_{2.5}$ from the polluted SCB region was transported out from the gap between the eastern TP and northern YGP edge.

### 3.5 Contribution of local emission and outflow transport

Local emissions and regional transport of air pollutants are the two key factors that affect air quality. The SCB region in the northeastern part of Southwest China, characterized by a deep-bowl

structure, is isolated by plateaus (TP in the west and YGP in the south) and mountains with a clean atmospheric environment. Haze pollution events with extremely high $PM_{2.5}$ concentrations over the SCB are ascribed to the accumulation of local anthropogenic pollutants and air pollutant transport over the basin (Wang et al., 2018; Qiao et al., 2019; Zhao et al., 2019). High local anthropogenic emissions

in the SCB dominate regional air pollution over the SCB (Liao et al., 2017). The transport of air pollutants from neighboring countries in South Asia is mostly concentrated in the neighboring regions of the southern TP and southern YGP (Wang et al., 2018; Zhao et al., 2019; Yin et al., 2020). Therefore, the anthropogenic emission data of South Asian neighboring countries of China are not included in the WRF-Chem simulation on haze pollution over SCB during 2–7 January 2017, considering the fewer effects of northward cross-border transport of air pollutants from    South Asian neighboring countries on air pollution in SCB with prevailing northeasterly wind during Asian winter monsoon season with a negligible contribution to the wintertime heavy haze pollution over the SCB region. Here, the differences in $PM_{2.5}$ concentrations between the numerical experiments, Emi-Real, and Emi-Non were analyzed to assess the contribution of regional air pollutant emissions to surface $PM_{2.5}$ concentrations in SCB and the impact of $PM_{2.5}$ transport from SCB to the surrounding plateaus and mountains.

Figure 11 shows the $PM_{2.5}$ concentrations originating from local emissions of primary $PM_{2.5}$, gaseous precursors of $PM_{2.5}$ over SCB, and the relative contribution rates to air pollution changes. The SCB's regional air pollutant emissions provided surface $PM_{2.5}$ from 40.6 to 136.2 µg m$^{-3}$, contributing 75.4–94.6% of total concentrations for the heavy pollution episode over SCB. This indicates the dominant role of local air pollutant emissions on air quality changes over this isolated deep basin in Southwest China. The surface $PM_{2.5}$ concentrations sourced from the regional air pollutant emissions over SCB were averaged, with 88.64, 91.04, and 65.96 µg m$^{-3}$ for P1, P2, and P3, respectively. However, interestingly, the average contribution rates of regional air pollutant emissions to surface $PM_{2.5}$ concentrations in SCB actually decreased from 90.7% in P1 to 85.6% in P2 and 83.3 % in P3 (Fig. 11). This could be attributed to the exchanges between the $PM_{2.5}$-rich airmass over SCB and $PM_{2.5}$-poor airmass in the surrounding plateaus and mountains over Southwest China (Figs. 8 and 9).

To assess the impact of the $PM_{2.5}$ transport from SCB on the air quality over the surrounding areas in Southwest China, we calculated the contribution and rates of SCB's regional air pollutant emissions to the $PM_{2.5}$ concentrations in the adjoining regions in the plateaus and mountains, based on the differences in $PM_{2.5}$ concentrations between Emi-Real and Emi-Non (Table 5). The near-surface prevailing northeasterly winds of the SCB brought $PM_{2.5}$ from the SCB to the eastern TP edge, the northern YGP edge, and the DBM region (Fig. 7), resulting in an increase in the concentrations of surface $PM_{2.5}$ during the heavy haze pollution event, with averages of 18.0, 31.3, and 10.4 µg m$^{-3}$,

respectively (Table 5). TP and YGP, the cleaner regions in China (Song et al., 2017; Zhan et al., 2018), received significant pollution due to the PM$_{2.5}$ transport from SCB. During the dissipation period of the heavy air pollution episode, the eastern TP edge and northern YGP regions gained peak imports of PM$_{2.5}$ at 22.9 and 41.9 μg m$^{-3}$ (Table 5). Thus, in this case, the downwind adjoining TP and YGP regions was the main receptor area of the SCB emissions.

Finally, the PM$_{2.5}$ contribution rates, i.e., the percentage of the PM$_{2.5}$ concentrations transported from the basin to those in the adjacent regions of plateaus and mountains were calculated for different periods of the heavy PM$_{2.5}$ pollution event over the SCB. The surface PM$_{2.5}$, in the eastern TP edge mostly originated from the source region of the SCB, with dominant contribution rates of 63.6 %, 67.4 %, and 72.7 % in the formation, maintenance, and dispersion periods, respectively. The PM$_{2.5}$ import from the SCB pollutant emissions also contributed to the majority of surface PM$_{2.5}$ concentrations in the northern YGP, with contribution rates of 58.3 %, 52.8 %, and 70.5 % during the three periods, with an overall contribution rate of 58.5% (averaged for the entire SCB heavy air pollution event). In contrast, the DBM region was less influenced by the SCB's emission sources, with a contribution rate of 31.0% (averaged during the heavy air pollution event).

## 4. Conclusions

By using the multiple ground observations, meteorological sounding data and micro pulse lidar retrievals as well as conducting modeling experiments with the WRF-Chem model, this study investigated the three-dimensional structures and the development mechanisms of the PM$_{2.5}$ for a wintertime heavy haze pollution event over SCB, an isolated deep basin in Southwest China. The roles of the basin pollutant emissions and the unique basin circulations were evaluated for their contributions to the 3D distribution of PM$_{2.5}$ over SCB and to the neighboring YGP, TP, and DBM regions.

The vertical structure of PM$_{2.5}$ in the lower troposphere over the SCB was characterized by unique hollows located between a high PM$_{2.5}$ layer at a height of 1.5–3 km and a high PM$_{2.5}$ surface layer. The hollow was developed by the interaction of the upper-level free tropospheric circulations and the lower-level topographic boundary layer. The southwesterlies passing over the TP and YGP resulted in a lee vortex over the SCB, which helped form and maintain high PM$_{2.5}$ concentrations, with well-developed vertical secondary circulations along the eastern TP upslope, whereas the

southwesterlies with the underlying high $PM_{2.5}$ layers were elevated in the dissipation of heavy $PM_{2.5}$ pollution over the SCB.

Due to the joint impact of the urban anthropogenic air pollutant emissions and the large terrain blocking flow at the eastern TP slope and YGP, high surface $PM_{2.5}$ concentrations were mostly

distributed in the northwestern and southern SCB regions. The tropospheric circulations, which altered the vertical diffusion of $PM_{2.5}$, exerted a strong impact on $PM_{2.5}$ distribution in the lower free troposphere. The high $PM_{2.5}$ centers in the lower free troposphere were distributed over the northwestern and southwestern SCB edges, as well as the central SCB regions. Driven by strong northeasterly winds in the dissipation period, $PM_{2.5}$, in the lower free troposphere converged to the west

boundary of the SCB and then transported to the eastern TP edge and the northern YGP edge areas.

The regional emissions of air pollutants in the SCB played a dominant role in the formation of heavy air pollution, contributing 75.4–94.6 % to surface $PM_{2.5}$ concentrations over the basin for the heavy pollution event studied herein. Furthermore, the surface $PM_{2.5}$ concentrations in the eastern TP were mostly transported from the SCB's emission sources, with contribution rates of 63.6 %, 67.4 %,

and 72.7 % for P1, P2, and P3, respectively. Similarly, the SCB also contributed the majority of surface $PM_{2.5}$ concentrations in the adjacent northern YGP, with an average contribution rate of 58.5 % for the whole SCB pollution period and a very high contribution of 70.5 % during the dissipation period. Therefore, the SCB region is the major source of air pollutants for the downwind receptor areas over the adjoining TP and YGP regions and affects the atmospheric environmental changes in Southwest

China.

This study highlights the unique and important three-dimensional structures of $PM_{2.5}$ and investigated their formation mechanisms and downwind outflow transport over the SCB. The deep basin terrain along with the TP and YGP forcing effect creates very complex $PM_{2.5}$ pollution conditions over the SCB region, which is significantly different from those over relatively flat regions. To

generalize our findings, further work with more case studies and regional climatic analyses with long-term observation data and numerical modeling with data assimilation and refined physical and chemical schemes are required. MEIC 2017 was not available for the WRF-Chem model. The SCB is located in Southwest China, with larger uncertainties in the anthropogenic emission inventory

compared to Eastern China. An accurate emission inventory could improve air pollution simulations

and air quality change assessments in future studies. Furthermore, as pointed out in this study, the PM$_{2.5}$

emission sources in the SCB greatly influence the regional environmental changes in Southwest China.

Thus, the regional transport modeling of air pollutants with careful consideration of the thermal and

dynamic forcing of the underlying complex plateau terrain should be further investigated.

*Data availability*. Data used in this paper can be provided by Zhuozhi Shu (shuzhuozhi@foxmail.com)

upon request.

*Author contributions*. ZS and TZ conducted the study design. ZS, TZ, JX, CW and LC conducted the

vertical observational experiment. ZS wrote the manuscript with the help of TZ and YL. LZ and YZ

assisted with data processing. HL and LS were involved in the scientific interpretation and discussion.

LL and YL provided the surface meteorological data. All of the authors provided commentary on the

paper.

*Competing interests*. The authors declare that they have no conflicts of interest.


*Acknowledgements*. This research was supported by the Second Tibetan Plateau Scientific Expedition

and Research (STEP) program (2019QZKK0105), the National Natural Science Foundation of China

(91744209, 91644223 and 91544109), the National Key R & D Program Pilot Projects of China

(2016YFC0203304) and Graduate Research and Innovation Projects of Jiangsu Province

(SJKE19_0941).

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

**Table 1**. Names of 18 observation sites with the corresponding site number (Fig. 1b) in the SCB.

| Number | 1 | 2 | 3 | 4 | 5 | 6 |
|--------|---|---|---|---|---|---|
| Names | Chengdu | Chongqing | Deyang | Guang'an | Leshan | Meishan |
| Number | 7 | 8 | 9 | 10 | 11 | 12 |
| Names | Mianyang | Nanchong | Neijiang | Suining | Yibin | Ziyang |
| Number | 13 | 14 | 15 | 16 | 17 | 18 |
| Names | Zigong | Luzhou | Ya'an | Bazhong | Dazhou | Guangyuan |

675                                    **Table 2**. Setting of physical and chemistry schemes in the WRF-Chem simulations

| | |
|--|--|
| Microphysics | Morrison 2-mom |
| Boundary layer | MYJ |
| Longwave radiation | RRTM |
| Shortwave radiation | RRTMG |
| Land surface | Noah |
| Cumulus convection | Grell 3D (none in D3) |
| Urban scheme | Single-layer |
| Chemistry | RADM2 |
| Aerosol particles | MADE/SORGAM |
| Photolysis | Madronich (TUV) |


**Table 3**. Statistical metrics of comparisons between simulated (Sim.) and observed (Obs.) 2-m air temperature (T2), surface relative humidity (RH) and 10-m wind speed (WS10) with the correlation

coefficient (R), mean bias(MB), mean error (ME) and root mean squared error (RMSE) during air pollution process over 2–7 January 2017.

| | Obs. | Sim. | R | MB | ME | RMSE |
|---|---|---|---|---|---|---|
| T2 | 9.9 ℃ | 9.2 ℃ | 0.78** | −0.7 | 1.7 | 2.1 |
| RH | 85.1 % | 77.7 % | 0.67** | −7.4 | 11.2 | 13.4 |
| WS10 | 1.2 m s$^{-1}$ | 1.5 m s$^{-1}$ | 0.41* | 0.3 | 0.8 | 1.1 |

Note: MB, ME, RMSE were calculated as following: $MB = \frac{1}{N}\sum_{i=1}^{N}(M_i - O_i)$; $ME = \frac{1}{N}\sum_{i=1}^{N}|M_i - O_i|$;

$RMSE = \sqrt{\frac{1}{N}\sum_{i=1}^{N}(M_i - O_i)^2}$; (M and O represented the results from simulation and observation). The

** and * respectively indicated the correlation coefficients R passing the 99% and 95% significant test.


**Table 4**. Statistical metrics of comparisons between simulated and observed surface PM$_{2.5}$

concentrations in two levels of light and heavy PM$_{2.5}$ pollution during 2–7 January 2017.

| | NMB (%) | NME (%) | MFB (%) | MFE (%) |
|---|---|---|---|---|
| Light pollution | −4.3 | 25.4 | −7.7 | 30.0 |
| Heavy pollution | −13.5 | 33.4 | −16.3 | 37.4 |

Note: NMB, NME, MFB and MFE were calculated as following: $NMB = \frac{\sum_{i=1}^{N}(M_i - O_i)}{\sum_{i=1}^{N}O_i}\cdot 100\%$; $NME =$

$\frac{\sum_{i=1}^{N}|M_i - O_i|}{\sum_{i=1}^{N}O_i}\cdot 100\%$; $MFB = \frac{1}{N}(2\cdot\frac{M_i - O_i}{M_i + O_i})\cdot 100\%$; $MFE = \sum_{i=1}^{N}\left|2\cdot\frac{M_i - O_i}{M_i + O_i}\right|\cdot 100\%$.


**Table 5**. Amounts and contribution rates of $PM_{2.5}$ trans-boundary transport from the SCB to surface

$PM_{2.5}$ concentrations averaged over the eastern TP edge (ETP), northern YGP edge (YGP) and DBM

region during the formation (P1), maintenance (P2) and dissipation (P3) periods of the heavy haze

pollution events over the SCB region.

|  | Region | P1 | P2 | P3 | Averages |
|---|---|---|---|---|---|
| Transport amount | ETP | 15.4 | 18.8 | 22.5 | 18.0 |
| ($\mu g\ m^{-3}$) | YGP | 30.1 | 27.5 | 41.9 | 31.3 |
|  | DBM | 8.6 | 13.5 | 8.4 | 10.4 |
| Contribution rates (%) | ETP | 63.6 | 67.4 | 72.7 | 66.6 |
|  | YGP | 58.3 | 52.8 | 70.5 | 58.5 |
|  | DBM | 26.7 | 36.6 | 30.1 | 31.0 |

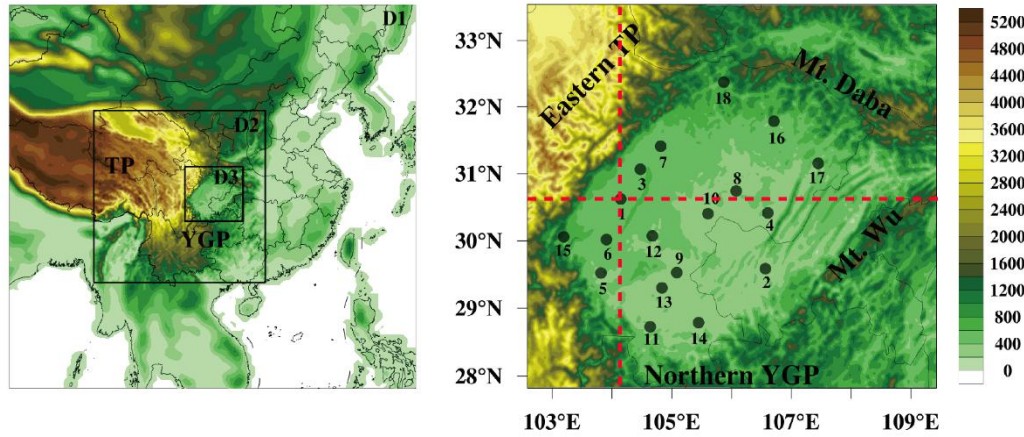


**Figure 1**. (Left panel) three nesting domains D1, D2 and D3 of WRF-Chem simulation with the terrain

heights (m in a.s.l.) and (right panel) the location of 18 urban observation sites (black dots, Table 1)

including site 1 (Chengdu) with the intensive sounding observations and site 15 (Ya'an) with the

ground-based MPL detection in the SCB with the surrounding Tibetan Plateau (TP), Yunnan-Guizhou

Plateau (YGP), Mountains Daba (Mt. Daba) and Wu (Mt. Wu) in Southwest China. The red dash lines

indicate the location of the cross sections respectively along 30.67º N and 104.02º E.

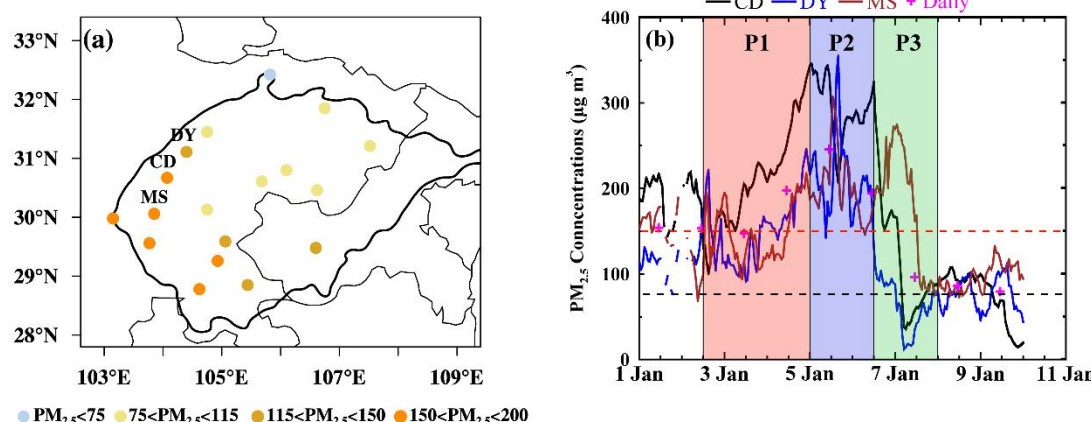

**Figure 2**. (a) Surface PM$_{2.5}$ concentrations over 18 urban sites averaged during the heavy haze

pollution over 2–7 January 2017, (b) hourly variations of PM$_{2.5}$ concentrations observed at the three

heaviest polluted cities Chengdu (CD), Deyang (DY), and Meishan (MS) (Fig. 1; Table 1) over 1–10

January 2017. The P1, P2 and P3 indicated respectively the formation, maintenance and dissipation

periods of heavy haze pollution with the light pollution level of 75 μg m$^{-3}$ (black dashed line) and

heavy pollution level of 150 μg m$^{-3}$ (red dashed line).

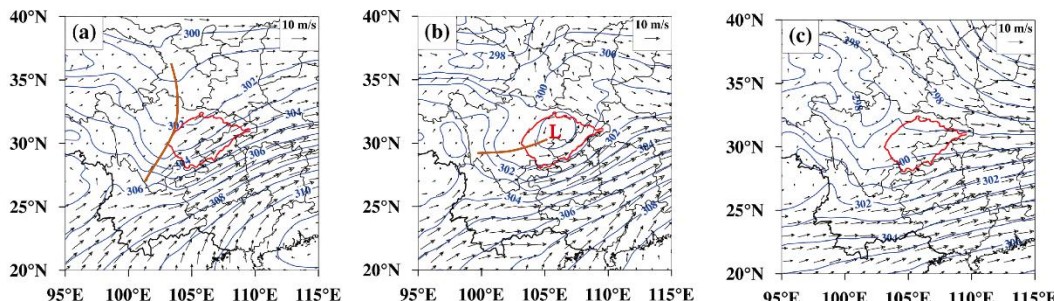

**Figure 3**. The 700 hPa geopotential height fields and wind vectors averaged during (a) P1, (b) P2 and

(c) P3 stages with the trough line (brown line) and low-pressure center (L). The SCB was outlined with

the red solid lines.

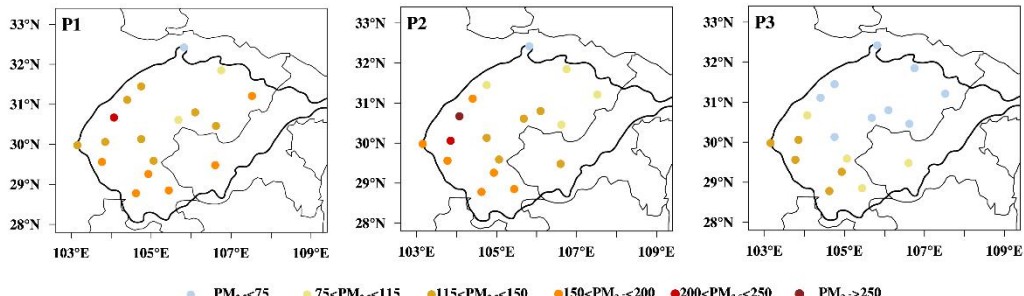

**Figure 4**. Spatial distributions of observed surface PM$_{2.5}$ concentrations in the SCB averaged in the

formation, maintenance and dissipation periods, P1, P2 and P3 (Fig. 2b), respectively, of the heavy

haze pollution episode over 2–7 January 2017.

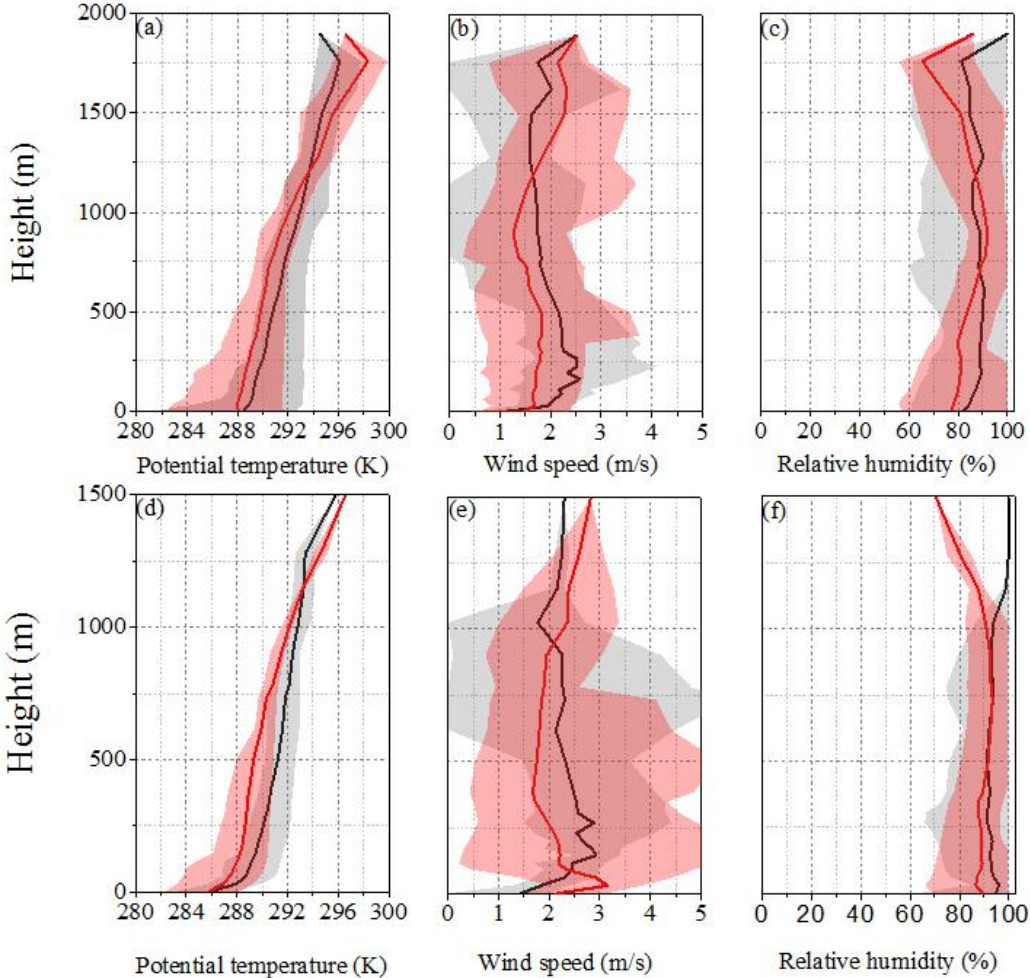

**Figure 5**. Comparisons of observed and simulated vertical profiles of potential temperature, wind speed, and relative humidity in the daytime (a, b, c) and nighttime (d, e, f) at Chengdu (site 1 in Fig. 1) during 2–7 January 2017. The red and gray shaded areas represented the variation range of simulation and observation in all vertical profiles with averaged values (lines), respectively.

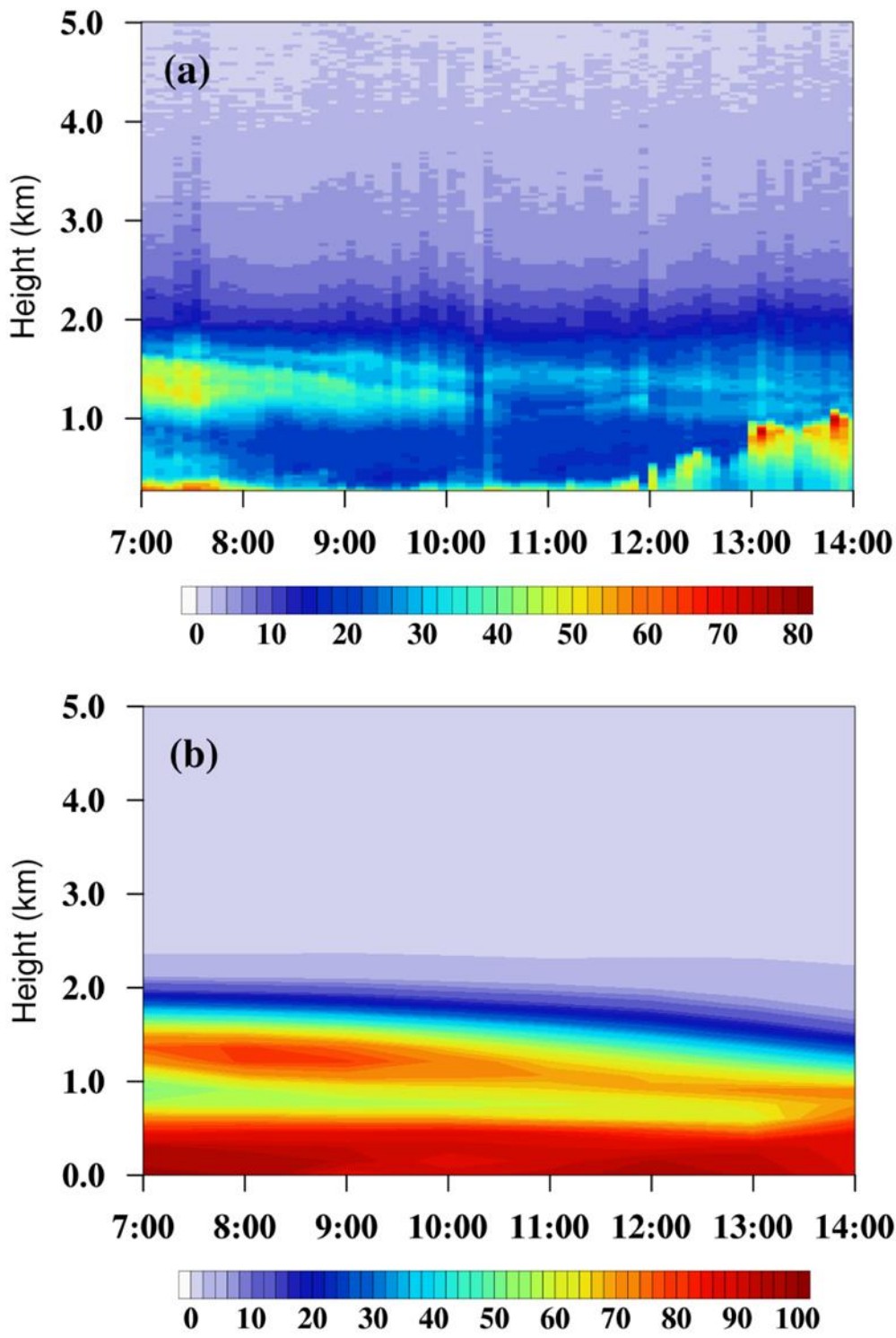

**Figure 6**. Vertical and time cross-sections of $PM_{2.5}$ mass concentrations ($\mu g\ m^{-3}$) from (a) MPL-4B

retrievals products and (b) simulation results at site 15 (Fig. 1; Table 1) on the western SCB edge

during 7:00 a.m.–2:00 p.m. on 5 January 2017.

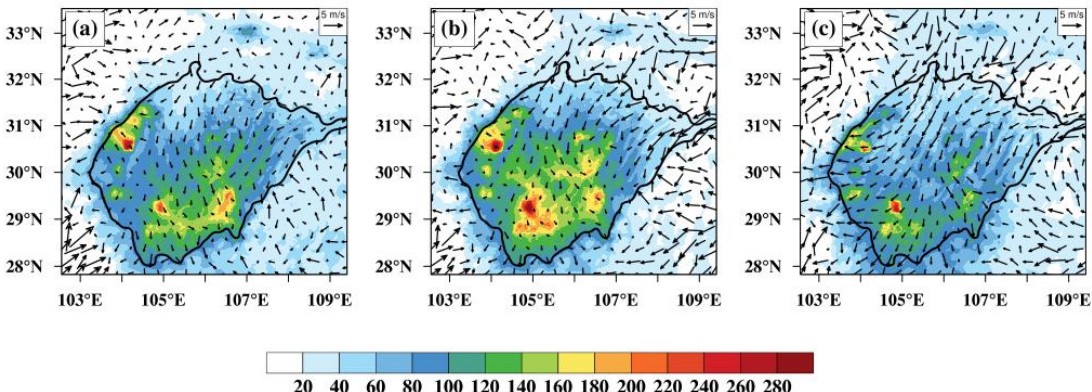

**Figure 7**. Horizontal distribution of surface PM$_{2.5}$ concentrations (μg m$^{-3}$; color contours) and wind

vectors at 10 m averaged in the periods (a) P1, (b) P2 and (c) P3, respectively. The SCB was outlined

with an altitude contour line of 750 m (a.s.l.; black lines).

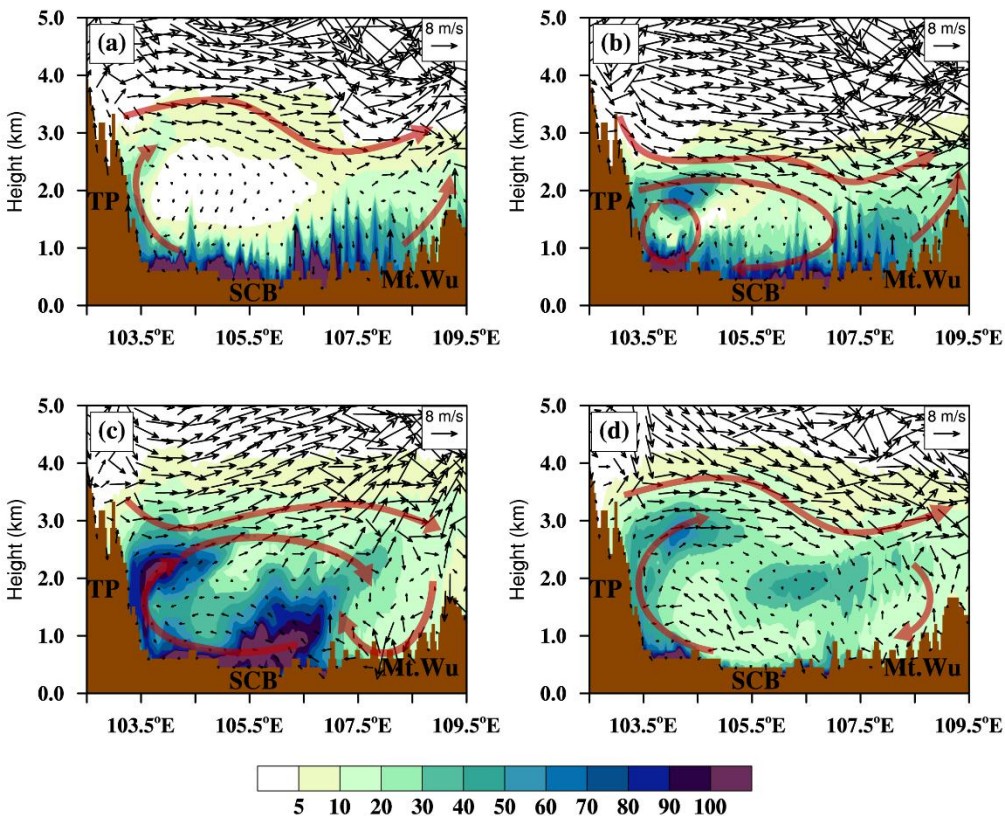


**Figure 8**. Height-longitude cross-sections of PM$_{2.5}$ concentrations (color contours: μg m$^{-3}$) and wind

vectors along 30.67° N in the (a) clean environment at 12:00 a.m. on 2 January 2017 (b) heavy air

pollution formation stage at 12:00 a.m. on 3 January 2017 (c) maintenance stage at 8:00 a.m.on 6

January 2017, and (d) dissipation stage at 8:00 a.m. on 7 January 2017. The brown arrows highlighted

the major air flows (red arrows) associated with the terrain of TP, SCB and Mt. Wu (filled brown

color).

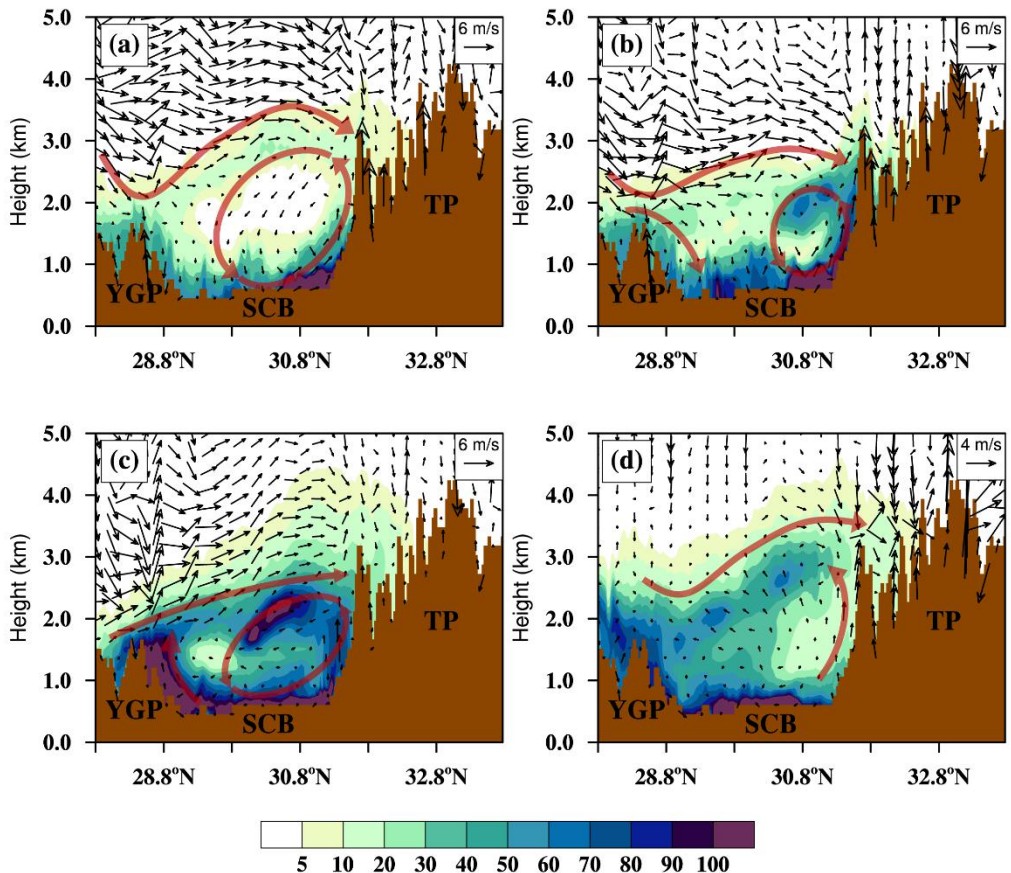

**Figure 9**. Same as Fig. 8, but for height-latitude cross-sections of PM$_{2.5}$ concentrations and wind

vectors.

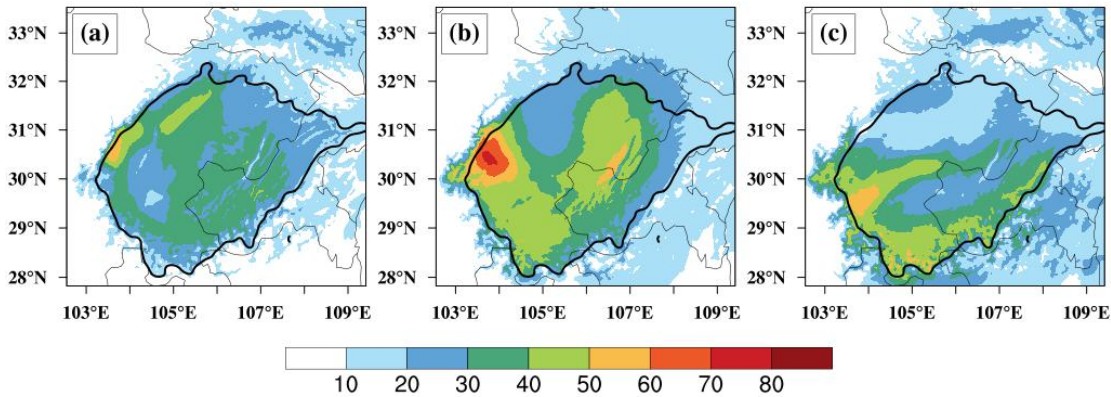

**Figure 10**. Horizontal distribution of PM$_{2.5}$ concentrations (color contours: μg m$^{-3}$) averaged between

1.5 and 2.5 km heights (in the lower free troposphere) for (a) formation, (b) maintenance and (c)

dissipation periods of heavy haze pollution event over the SCB. The SCB was outlined with an altitude

contour of 750 m in a.s.l. (dark black lines).

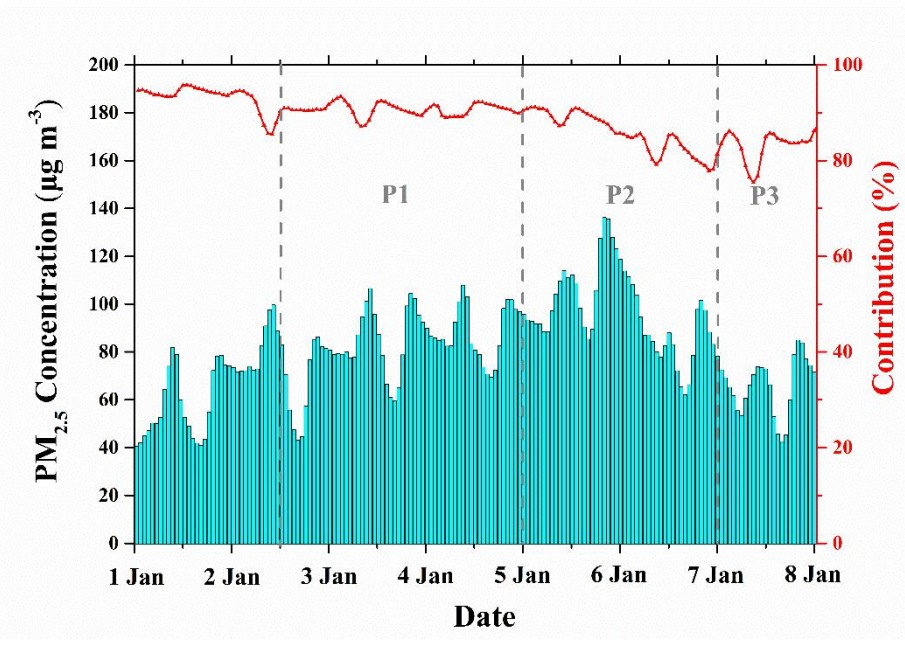


**Figure 11**. Hourly variations of surface PM2.5 concentrations originating from the SCB's anthropogenic

emissions (blue filled areas) and the contribution proportions to the basin surface PM2.5 levels (red

curve) during 1–7 January 2017 based on the regional averages over the SCB.