# Peer review of "Elevated 3D structures of PM2.5 and impact of complex terrain-forcing circulations on heavy haze pollution over Sichuan Basin, China"

_Atmospheric Chemistry and Physics, 2020_

## Referee Comment (RC1) · Anonymous Referee #1 · 2 Jan 2021

Review of the manuscript Number: acp-2020-1161

Elevated 3D structures of PM2.5 and impact of complex terrain-forcing circulations on heavy haze pollution over Sichuan Basin, China

submitted for publication to Atmospheric Chemistry and Physics

General Comment

The paper analyzes an episode with high concentrations of PM2.5 in the Sichuan Basin (China), combining observations and numerical simulations. The paper is potentially interesting, in particular for the peculiar interaction between meso and local circulations

and pollutant emissions, which leads to the formation of an elevated pollutant layer. However, the discussion of the results should be improved before the paper can be accepted for publication.

Specific Comments

1) Meteorological conditions

I. a general meteorological overview of the event, including a synoptic characterization, is missing in the paper.

2) Model set-up

I. The Authors adopt a grid ratio of 1:4, while an odd grid ratio is recommended because for even values interpolation errors arise due to the nature of Arakawa C-grid staggering. The Authors should at least discuss this choice.

II. The Authors say that the "vertical turbulent diffusion coefficient of the boundary layer was reduced". This aspect should be better discussed, since it might significantly affect the results.

III. No information about the vertical discretization is given. An adequate vertical resolution is fundamental to evaluate the thermal stratification over complex terrain.

3) Model validation

I. The Authors propose a series of statistical indexes for evaluating model results, both for meteorological variables and PM2.5. From these statistical indexes it is difficult to judge the performance of the model, regarding in particular the time evolution of observed and simulated variables. I strongly suggest to show some representative time series to better evaluate the model performance at some representative location.

II. Figure 4 presents a comparison between the vertical profiles of potential temperature, wind speed and relative humidity from observations and model results. Also in this case it is difficult to evaluate model results, since only mean profiles and the variation

range over the entire period are presented. I suggest to show also some representative profiles at some specific hours. In particular, it would be interesting to evaluate how the WRF model is able to capture the vertical temperature profile, since atmospheric stability is crucial for pollutant dispersion. In many points in the paper a temperature inversion is cited, but the simulation of this temperature inversion is never discussed. For example, at lines 243-250, "thermo-dynamical structures" and "stable stratification" are cited, but, without a representative figure, it is difficult to follow the discussion of the results.

4) Language

I. Although the paper is rather well written, a review by a native English speaker would be beneficial

Minor and technical remarks

Page 2, line 79: "Section 2 introduced...". Here and in other parts of the paper I would use the present tense (when referring to tables, figures...).

Figures 7 and 8: the location of the cross sections should be indicated in Fig. 1.

---

## Referee Comment (RC2) · Anonymous Referee #2 · 6 Jan 2021

\*\*\*\*\*\*\*\*\*\*\*\*\*\*\*\*\*\*\*\*\*\*\*\*\*\*\*\*

General Comment

\*\*\*\*\*\*\*\*\*\*\*\*\*\*\*\*\*\*\*\*\*\*\*\*\*\*\*\*

This paper analyzed the three-dimensional distribution of PM2.5 concentrations in Sichuan Basin during a heavy haze pollution episode in January 2017. The topic is quite interesting; However, many discussions are only general descriptions of phenomena and processes, lacking in-depth analysis and discussion. This makes the article as a whole difficult to follow.

[Figure]
* * *
Specific Comments
* * *
Line 115

The Multi-resolution Emission Inventory for China (MEIC) has been updated to 2017 (http://www.meicmodel.org), while the authors used the data of 2012 for the simulation period in January, 2017. The author should explain the mismatch.

Further, the first domain (D1) of the study area includes China and its neighboring countries/regions. In this section, the author only stated that the anthropogenic emission sources used in their study is MEIC data, but as far as I know, MEIC data only includes the anthropogenic emission sources in China, while the emissions from natural sources and neighboring countries/regions are not included. How did the author consider this in their simulation process? If the emission data of neighboring countries/regions are not included, there will be great uncertainty about the results of the section 3.5 (Contribution of local emission and outflow transport), because the surrounding emissions are ignored.

Line 157

3.1 Model evaluation As we know, China has adopted active pollution source control policies in the last 5 years, and the intensity, the temporal and spatial distribution of emission sources will vary greatly from year to year. The author selected the 2012 MEIC inventory as its emission data. Thus the model evaluation result may not be convincing.

Line 216

To examine the vertical structures of PM2.5 concentrations over SCB, we selected the urban site 1 (104.02° E; 30.67° N) in Chengdu (cf. Fig. 1) as a reference point to

investigate the distributions of PM2.5 and the atmospheric circulations respectively in the vertical-meridional and vertical-zonal cross-sections.

Why do you select he urban site 1 (104.02° E; 30.67° N) in Chengdu for the vertical discussion. Do you have any special purpose? Chengdu is located in the far west side of the SCB, and other sites in the central area of SCB maybe are better choices, as the wind vectors shows in Figure 6.

Line 275

Figure 10 showed the PM2.5 concentrations emitted from the regional air pollutant sources over the SCB region and the relative contribution rates to air pollution changes. The expression here shows the author's conceptual misunderstanding of the source of PM2.5. How can the "PM2.5 concentrations" be "emitted"?

Line 278

The SCB's regional air pollutant emissions provided surface PM2.5 concentrations from 40.6 to 136.2 $\mu$g m-3, contributing 75.4–94.6 % of surface PM2.5 concentrations for the heavy pollution episode over SCB, indicating its dominant role over this isolated deep basin in Southwest China.

What does " indicating its dominant role over this isolated deep basin in Southwest China " mean? It is hard to follow.

Line 279

The regionally emitting PM2.5 concentrations averaged over SCB were 88.64, 91.04 and 65.96 $\mu$g m-3 for the formation, maintenance and dissipation periods, respectively. Same as above. How can the "PM2.5 concentrations" be "emitted"?

Line 284

We think the exchanges of PM2.5 between the polluted air over SCB and the cleaner environment air over the surrounding plateaus and mountains in Southwest China play

a role in this process. (Figs. 7 and 8).

How do you think the PM2.5 can be "exchanged" between the polluted air over SCB and the cleaner environment air?

Line 560

Table 5. Please give the definite range of the eastern TP edge (ETP), northern YGP edge (YGP) and DBM region.

Line 600

Figure 5(a). Why only 8 hours data are presented here? There is an abnormal value around half past 10 a.m., please give the reason.

Line 610 -615

Figure 8. Does this cross section along 104.02° E? Please specify. Same as previous mentioned, why do you select this site 1 (104.02° E; 30.67° N)? Do you think it may be a better choice if you put the cross section along the wind vectors from northeast to southwest?

Line 625

Figure 10. How the values of surface PM2.5 concentrations are calculated? The regional average of the SCB or the average of several monitoring sites in SCB?

---

## Author Comment (AC1) · 18 Jan 2021

Dear Editors and Referees:

Thank you very much for your review and comments concerning our manuscript entitled "Elevated 3D structures of PM$_{2.5}$ and impact of complex terrain-forcing circulations on heavy haze pollution over Sichuan Basin, China" [MS No.: acp-2020-1161]. Those comments are all valuable and helpful for revising and improving the manuscript. We have studied comments carefully and have accordingly made the revisions. Revised parts are highlighted with Track Changes in the revised manuscript. In the following we quoted each review question in the square brackets and added our response after each paragraph.

===================================================

**Response to Referee #1**

===================================================

*[The paper analyzes an episode with high concentrations of PM2.5 in the Sichuan Basin (China), combining observations and numerical simulations. The paper is potentially interesting, in particular for the peculiar interaction between meso and local circulations and pollutant emissions, which leads to the formation of an elevated pollutant layer. However, the discussion of the results should be improved before the paper can be accepted for publication.]*

**Response 1:** Many thanks for your encouraging comments. We have revised the manuscript accordingly. All the revisions have been highlighted with Track Changes in the revised manuscript. The point-by-point responses to the reviewer's comments are as follows.

*General comments*

*[1. A general meteorological overview of the event, including a synoptic characterization, is missing in the paper.]*

**Response 2:** Following the referee's comment, We have plotted the 700hPa geopotential heights and wind vectors in three stages (newly added Figure 3) over SCB and the surrounding regions with the meteorology analysis data of ERA-Interim.

We also characterized the overview of the synoptic conditions in the revised Sect. 2.4 as follows:

"The meteorological overview of the haze event was characterized with the 700-hPa fields of geopotential heights and wind vectors (Fig. 3). A trough in the mid-latitude westerlies moved eastwards from the eastern edge of TP to the western SBC margin during the haze formation stage P1, the trough of low pressure was evolved over the SCB region during the haze maintenance stage P2, and the westerly trough shifted out the SCB region with the low-pressure system disappearing in the haze dissipation stage P3 (Fig. 3). The changes of atmospheric circulations in the formation, maintenance and dissipation stages reflected the meteorological modulation on heavy haze development over SCB in associated with the effect of TP topography on the westerlies. "

[Figure]

**Figure 3**. The 700 hPa geopotential height fields and wind vectors averaged during (a) P1, (b) P2 and (c) P3 stages with the trough line (brown line) and low-pressure center (L). The SCB was outlined with the red solid lines.

From the formation to the maintenance and the dissipation periods of haze pollution, the near-suface prevailing northeasterly winds strengthened gradually over SCB. During the formation and maintenance stage, the surface winds were weak (1.4–1.7 m s$^{-1}$) over SCB, which was insufficient to dispel the air pollutants, but to continuously accumulate $PM_{2.5}$ locally from light to heavy pollution conditions (Fig. 7a, Fig. 7b). By the dissipation period, the northeasterly winds intensified and removed $PM_{2.5}$ from SCB (Fig. 7c).

*[2. The Authors adopt a grid ratio of 1:4, while an odd grid ratio is recommended because for even values interpolation errors arise due to the nature of Arakawa C-grid staggering. The authors should at least discuss this choice.]*

**Response 3:** Thanks for the referee's suggestion. In the revised manuscript, we have accordingly added the following discussions in the revised Sect. 2.3:

"Considering the complex terrain underlying over the SCB's deep basin and surrounding plateaus and mountains in Southwest China, we adopted a grid ratio of 1:4 for simulation experiments with precisely defined horizontal resolution in the study. It should be pointed out that the even grid ratio may cause the interpolation errors at the nested-domains boundary conditions due to the nature of Arakawa C-grid staggering."

*[3. The Authors say that the "vertical turbulent diffusion coefficient of the boundary layer was reduced". This aspect should be better discussed, since it might significantly affect the results.]*

**Response 4:** Following the referee's comments, we have added the below discussions into the revised Sect. 2.3.

"High $PM_{2.5}$ levels in the atmosphere could significantly reduce the near-ground solar radiation for the stable atmospheric stratification, which decreases vertical turbulent diffusion in the boundary layer (Wang et al., 2019), that is an important mechanism for severe haze pollution formation with the explosive growth of $PM_{2.5}$ (Zhong et al., 2018). The overestimated vertical diffusion capacity under poor air quality conditions (Ren et al., 2019) causes the deviation of air pollutants concentrations simulated in air quality model (Wang et al., 2018). In this study, the vertical turbulent diffusion coefficient of the atmospheric boundary layer was cut a

half down for better simulation of the 3D structures of $PM_{2.5}$ during the heavy air pollution over the SCB region."

**References**

Wang, L., Liu, J., Gao, Z., Li, Y., Huang, M., Fan, S., Zhang, X., Yang, Y., Miao, S., Zou, H., Sun, Y., Chen, Y., and Yang, T.: Vertical observations of the atmospheric boundary layer structure over Beijing urban area during air pollution episodes, Atmos. Chem. Phys., 19, 6949–6967, https://doi.org/10.5194/acp-19-6949-2019, 2019.

Zhong, J., Zhang, X., Dong, Y., Wang, Y., Liu, C., Wang, J., Zhang, Y., and Che, H.: Feedback effects of boundary-layer meteorological factors on cumulative explosive growth of $PM_{2.5}$ during winter heavy pollution episodes in Beijing from 2013 to 2016, Atmos. Chem. Phys., 18, 247–258, https://doi.org/10.5194/acp-18-247-2018, 2018.

Ren, Y., Zhang, H., Wei, W., Wu, B., Liu, J., Cai, X., and Song, Y.: Comparison of the turbulence structure during light and heavy haze pollution episodes, Atmospheric Research, 230, 0169-8095, https://doi.org/10.1016/j.atmosres.2019.104645, 2019.

Wang, H., Peng, Y., Zhang, X., Liu, H., Zhang, M., Che, H., Cheng, Y., and Zheng, Y.: Contributions to the explosive growth of $PM_{2.5}$ mass due to aerosol–radiation feedback and decrease in turbulent diffusion during a red alert heavy haze in Beijing–Tianjin–Hebei, China, Atmos. Chem. Phys., 18, 17717–17733, https://doi.org/10.5194/acp-18-17717-2018, 2018.

*[4. No information about the vertical discretization is given. An adequate vertical resolution is fundamental to evaluate the thermal stratification over complex terrain.]*

**Response 5:** Thanks for the referee's comments. The information about the vertical discretization is added in the revised manuscript (lines 116-118) as follows:

"An adequate vertical resolution is fundamental to evaluate the thermal stratification over complex terrain. Therefore, 35 vertical layers were set with the fine resolutions of 30–120 m in the boundary layer."

*[5. The Authors propose a series of statistical indexes for evaluating model results, both for meteorological variables and PM2.5. From these statistical indexes it is difficult to judge the performance of the model, regarding in particular the time evolution of observed and simulated variables. I strongly suggest to show some representative time series to better evaluate the model performance at some representative location.]*

**Response 6:** Following the referee's suggestion, the hourly variations of PM$_{2.5}$ concentrations, 2 m air temperature, surface relative humidity and near-surface wind speed in Chengdu (site 1), Suining (site 10) and Zigong (site 13) were shown in Figures S1 and S2 in the supplement of manuscript. The comparisons between observation and simulation also were evaluated with the reasonable WRF-Chem modeling performance.

[Figure]

**Figure S1**. Hourly variations of observed (black curves) and simulated (red curves) PM$_{2.5}$ concentrations respectively in (a) Chengdu (site 1), (b) Suining (site 10) and (c) Zigong (site 13) during the haze pollution episode.

[Figure]

**Figure S2**. Hourly variations of observed (black curves) and simulated (red curves) 2 m air temperature, surface relative humidity and wind speed results respectively in (a) Chengdu (site 1), (b) Suining (site 10) and (c) Zigong (site 13) during the haze pollution episode in 2017.

*[6. Figure 4 presents a comparison between the vertical profiles of potential temperature, wind speed and relative humidity from observations and model results. Also in this case it is difficult to evaluate model results, since only mean profiles and the variation range over the entire period are presented. I suggest to show also some representative profiles at some specific hours. In particular, it would be interesting to evaluate how the WRF model is able to capture the vertical temperature profile, since atmospheric stability is crucial for pollutant dispersion. In many points in the paper a temperature inversion is cited, but the simulation of this temperature inversion is*

*never discussed. For example, at lines 243-250, "thermo-dynamical structures" and "stable stratification" are cited, but, without a representative figure, it is difficult to follow the discussion of the results.]*

**Response 7:** Following the referee's suggestions, the vertical air temperature profiles were evaluated with the comparisons of vertical air temperature profiles between observation and simulation during different haze periods at (a) 11:00 p.m. on 2 January, (b) 05:00 a.m. on 3 January, (c) 02:00 p.m. on 3 January, (d) 02:00 p.m. on 4 January, (e) 08:00 p.m. on 4 January, (f) 05:00 p.m. on 5 January, (g) 08:00 a.m. on 6 January, (h) 11:00 a.m. on 6 January and (i) 11:00 a.m. on 7 January 2017. (Figure S3 in the supplement of manuscript) with the following description added in the revised Sect. 3.1:

"Comparing with the observed air temperature, the WRF-Chem simulations were evaluated to reasonably capture the vertical temperature profiles for understanding atmospheric stability in the vertical thermo-dynamical structures in the boundary layer over SCB (Fig. S3)."

We have clarified the description of "thermo-dynamical structures" and "stable stratification" into the revised Sect. 3.3.

"The potential temperature vertical gradients (Fig. 5), which is used for assessing atmospheric stability, were estimated respectively with 4.0 K/km, 7.8 K/km and 5.2 K/km in the boundary layer during the formation, maintenance and the dissipation periods of haze pollution with near-surface strong temperature inversion (Fig. S3), presenting the thermo-dynamical structure with stable stratification in the atmospheric boundary layer weakening air pollutant dispersion."

[Figure]

**Figure S3**. Comparisons of vertical profiles of air temperature between observation (black curves) and simulation (red curves) at (a) 11:00 p.m. on 2 January, (b) 05:00 a.m. on 3 January, (c) 02:00 p.m. on 3 January, (d) 02:00 p.m. on 4 January, (e) 08:00 p.m. on 4 January, (f) 05:00 p.m. on 5 January, (g) 08:00 a.m. on 6 January, (h) 11:00 a.m. on 6 January and (i) 11:00 a.m. on 7 January 2017.

*[7. Although the paper is rather well written, a review by a native English speaker would be beneficial.]*

**Response 8:** Thanks for your positive comments. A native English speaker has reviewed the paper to improve the language.

*Specific comments:*

*[Page 2, line 79: "Section 2 introduced…". Here and in other parts of the paper I would use the present tense (when referring to tables, figures…).]*

**Response 9:** Thanks for the careful review, we have changed with the present tense (when referring to tables, figures…).

*[Figures 7 and 8: the location of the cross sections should be indicated in Fig. 1.]*

Response 10: We have indicated the location of the cross sections in the modified Figure 1.

[Figure]

**Figure 1**. (Left panel) three nesting domains D1, D2 and D3 of WRF-Chem simulation with the terrain heights (m in a.s.l.) and (right panel) the location of 18 urban observation sites (black dots, Table 1) including site 1 (Chengdu) with the intensive sounding observations and site 15 (Ya'an) with the ground-based MPL detection in SCB with the surrounding Tibetan Plateau (TP), Yunnan-Guizhou Plateau (YGP), Mountains Daba (Mt. Daba) and Wu (Mt. Wu) in Southwest China. The red dash lines indicate the location of the cross sections respectively along 30.67º N and 104.02º E.

---

## Author Comment (AC2) · 18 Jan 2021

Dear Editors and Referees:

Thank you very much for your review and comments concerning our manuscript entitled "Elevated 3D structures of PM$_{2.5}$ and impact of complex terrain-forcing circulations on heavy haze pollution over Sichuan Basin, China" [MS No.: acp-2020-1161]. Those comments are all valuable and helpful for revising and improving manuscript. We have studied comments carefully and have accordingly made the revisions. Revised parts are highlighted with Track Changes in the revised manuscript. In the following we quoted each review question in the square brackets and added our response after each paragraph.

==================================================

**Response to Referee #2**

==================================================

*[This paper analyzed the three-dimensional distribution of PM2.5 concentrations in Sichuan Basin during a heavy haze pollution episode in January 2017. The topic is quite interesting; However, many discussions are only general descriptions of phenomena and processes, lacking in-depth analysis and discussion. This makes the article as a whole difficult to follow.]*

**Response 1:** Many thanks for your encouraging comments. We have revised the manuscript accordingly with in-depth analysis and discussion. All the revisions have been highlighted with Track Changes in the revised manuscript. The point-by-point responses to the reviewer's comments are as follows.

*[1. Line 115: The Multi-resolution Emission Inventory for China (MEIC) has been updated to 2017 (http://www.meicmodel.org), while the authors used the data of 2012 for the simulation period in January, 2017. The author should explain the mismatch. Further, the first domain (D1) of the study area includes China and its neighboring countries/regions. In this section, the author only stated that the anthropogenic emission sources used in their study is MEIC data, but as far as I know, MEIC data only includes the anthropogenic emission sources in China, while the emissions from natural sources and neighboring countries/regions are not included. How did the author consider this in their simulation process? If the emission data of neighboring countries/regions are not included, there will be great uncertainty about the results of the section 3.5 (Contribution of local emission and outflow transport), because the surrounding emissions are ignored.]*

**Response 2:** Thanks for the comments. We have accordingly added the following explanation and discussions in the revised conclusions (Sect. 4.):

"The MEIC 2017 was not available for our WRF-Chem modelling experiments. The SCB is located in Southwest China with larger uncertainties in anthropogenic emission inventory comparing to Eastern China. The accurate emission inventory could improve air pollution simulations and air quality change assessments in further study."

Furthermore, we have clarified the emissions from natural sources and neighboring countries/regions in the revised manuscript (Sect. 3.5) as follows:

"The Model of Emissions of Gases and Aerosols from Nature (v2.1) was applied for the natural emission sources in the simulation with dust emission parameterization. The SCB in the northeastern part of Southwest China with a deep-bowl structure is isolated by the plateaus (TP in west and YGP in south) and mountains with a clean atmospheric environment. High local anthropogenic emissions in the SCB dominate the regional air pollution (Liao et al., 2017). The transport of air pollutants from neighboring countries/regions are mostly concentrated in the neighboring regions in the southern Tibetan Plateau and the southern Yunnan-Guizhou Plateau (Wang et al., 2018; Zhao et al., 2019; Yin et al., 2020). Therefore, the anthropogenic emission data of neighboring countries/regions are not included in the WRF-Chem simulation on haze pollution over SCB during 2-8 January, 2017, considering the less effects of air pollutant cross-border transport on wintertime air pollution in SCB with the ignorable contribution to the wintertime heavy haze pollution over the SCB region."

**References**

Liao, T., Wang, S., Ai, J., Gui, K., Duan, B., Zhao, Q., Zhang, X., Jiang, W., and Sun, Y.: Heavy pollution episodes, transport pathways and potential sources of $PM_{2.5}$ during the winter of 2013 in Chengdu (China), Science of the Total Environment, 584, 1056-1065,

https://doi.org/10.1016/j.scitotenv.2017.01.160, 2017.

Wang, H., Tian, M., Chen, Y., Shi, G., Liu, Y., Yang, F., Zhang, L., Deng, L., Yu, J., Peng, C., and Cao, X.: Seasonal characteristics, formation mechanisms and source origins of $PM_{2.5}$ in two megacities in Sichuan Basin, China, Atmos. Chem. Phys., 18, 865–881, https://doi.org/10.5194/acp-18-865-2018, 2018.

Zhao, S., Yu, Y., Qin, D., Yin, D., Dong, L., and He, J.: Analyses of regional pollution and transportation of $PM_{2.5}$ and ozone in the city clusters of Sichuan Basin, China, Atmospheric Pollution Research, 10(2), 374-385, https://doi.org/10.1016/j.apr.2018.08.014, 2019.

Yin, D., Zhao, S., Qu, J., Yu, Y., Kang, S., Ren, X., Zhang, J., Zou, Y., Dong, L., Li, J., He, J., Li, P., and Qin, D.: The vertical profiles of carbonaceous aerosols and key influencing factors during wintertime over western Sichuan Basin, China, Atmospheric Environment, 223, 1352-2310, https://doi.org/10.1016/j.atmosenv.2020.117269, 2020.

*[2. Line 157: 3.1 Model evaluation As we know, China has adopted active pollution source control policies in the last 5 years, and the intensity, the temporal and spatial distribution of emission sources will vary greatly from year to year. The author selected the 2012 MEIC inventory as its emission data. Thus the model evaluation result may not be convincing.]*

**Response 3:** We agree with the referee that China has adopted active pollution source control policies in the last 5 years, and the intensity, the temporal and spatial distribution of emission sources will vary greatly from year to year. We have added the explanation and discussions about the 2012 MEIC emission data in the revised conclusions as follows:

"The MEIC 2017 was not available for our WRF-Chem modelling experiments. The SCB is located in Southwest China with larger uncertainties in anthropogenic emission inventory comparing to Eastern China, The accurate emission inventory could improve air pollution simulations and air quality change assessments in further study."

*[3. Line 216: To examine the vertical structures of PM2.5 concentrations over SCB, we selected the urban site 1 (104.02◦ E; 30.67◦ N) in Chengdu (cf. Fig. 1) as a reference point to investigate the distributions of PM2.5 and the atmospheric circulations respectively in the vertical-meridional and vertical-zonal cross-sections. Why do you select the urban site 1 (104.02◦ E; 30.67◦ N) in Chengdu for the vertical discussion. Do you have any special purpose? Chengdu is located in the far west side of the SCB, and other sites in the central area of SCB maybe are better choices, as the wind vectors shows in Figure 6.]*

**Response 4:** We selected the urban site 1 (104.02° E; 30.67° N) in Chengdu for the vertical discussion with the following purposes:

1)   The terrain effect of TP, the "world roof" on the mid-latitude westerlies could modulate haze pollution in the downstream region over China (Xu et al., 2016). The SCB is immediately to the east of TP with a large elevation drop exceeding 3000 m over a short horizontal distance. The unique terrain effect generates the asymmetries of meteorological and air pollutant distribution (Zhang et al., 2019). To better understand the elevated 3D structures of $PM_{2.5}$ with the impact of TP terrain-forcing circulations over the pollution episode.

2)   Chengdu (site 1), is a metropolis in SCB with high anthropogenic emissions and the most polluted environment in Southwest China (Ning et al., 2018). It is important to investigate how the urban surface high $PM_{2.5}$ levels evolved vertically with the combination between the high urban emissions and TP's terrain-forcing lifting over the SCB.

Furthermore, we have plotted the cross-sections of $PM_{2.5}$ and wind vectors along the near-surface prevailing northeastern wind across the central SCB (blue line in Figure S4 of manuscript supplement). The vertical changes of $PM_{2.5}$ with the terrain-forcing local circulations by YGP-terrain effects were remarkably presented in the different stages of the heavy haze pollution event.

[Figure]

**Figure S4**. Northeast-southwest cross-sections along the near-surface prevailing wind (blue line) over SCB (left panel), PM$_{2.5}$ concentrations (color contours: μg m$^{-3}$) and wind vectors (right panel) in (a) the relative clean environment at 12:00 a.m. on 2 January, (b) heavy air pollution formation stage at 12:00 a.m. on 3 January and (c) maintenance stage at 8:00 a.m.on 6 January 6 and (d) dissipation stage at 8:00 a.m. on 7 January, 2017.

**References**

Xu, X., Zhao, T., Liu, F., Gong, S. L., Kristovich, D., Lu, C., Guo, Y., Cheng, X., Wang, Y., and Ding, G.: Climate modulation of the Tibetan Plateau on haze in China, Atmos. Chem. Phys., 16, 1365–1375, https://doi.org/10.5194/acp-16-1365-2016, 2016.

Zhang, L., Guo, X., Zhao, T., Gong, S., Xu, X., Li, Y., Luo, L., Gui, K., Wang, H., Zheng, Y, and Yin, X.: A modelling study of the terrain effects on haze pollution in the Sichuan Basin, Atmos. Environ., 196, 515 77–85, https://doi.org/10.1016/j.atmosenv.2018.10.007, 2019.

Ning, G., Wang, S., Ma, M., Ni, C., Shang, Z., Wang, J. and Li, J.: Characteristics of air pollution in different zones of Sichuan Basin, China, Sci. Total. Environ., 612, 975–984, 440 https://doi.org/10.1016/j.scitotenv.2017.08.205, 2018.

*[4. Line 275: Figure 10 showed the PM2.5 concentrations emitted from the regional air pollutant sources over the SCB region and the relative contribution rates to air pollution changes.*
*The expression here shows the author's conceptual misunderstanding of the source of PM2.5. How can the "PM2.5 concentrations" be "emitted"?]*

**Response 5:** We modified the expression as "Figure 11 shows the PM$_{2.5}$

concentrations originated from local emissions of primary $PM_{2.5}$ and gaseous precursors of $PM_{2.5}$ over SCB and the relative contribution rates to air pollution changes."

*[5. Line 278: The SCB's regional air pollutant emissions provided surface PM2.5 concentrations from 40.6 to 136.2 μg m-3, contributing 75.4–94.6 % of surface PM2.5 concentrations for the heavy pollution episode over SCB, indicating its dominant role over this isolated deep basin in Southwest China.*
*What does "indicating its dominant role over this isolated deep basin in Southwest China" mean? It is hard to follow.]*

**Response 6:** In the revised manuscript, we have clarified this sentence as "The SCB's regional air pollutant emissions provided surface $PM_{2.5}$ from 40.6 to 136.2 μg $m^{-3}$, contributing 75.4–94.6 % proportion of total concentrations for the heavy pollution episode over SCB, indicating the dominant role of local air pollutant emissions on air quality changes over this isolated deep basin in Southwest China."

*[6. Line 279: The regionally emitting PM2.5 concentrations averaged over SCB were 88.64, 91.04 and 65.96 μg m-3 for the formation, maintenance and dissipation periods, respectively.*
*Same as above. How can the "PM2.5 concentrations" be "emitted"?]*

**Response 7:** The expression has been corrected as "The surface $PM_{2.5}$ concentrations sourced from the regional air pollutant emissions over SCB were averaged respectively with 88.64, 91.04 and 65.96 μg $m^{-3}$ for the formation, maintenance and dissipation periods of air pollution."

*[7. Line 284: We think the exchanges of PM2.5 between the polluted air over SCB and the cleaner environment air over the surrounding plateaus and mountains in Southwest China play a role in this process. (Figs. 7 and 8).*
*How do you think the PM2.5 can be "exchanged" between the polluted air over SCB and the cleaner environment air?]*

**Response 8:** Thanks for the referee's careful review. The sentence has been modified as ", which could be attributed by the exchanges between the $PM_{2.5}$-rich airmass over

SCB and PM$_{2.5}$-poor airmass in the surrounding plateaus and mountains over Southwest China."

*[8. Line 560: Table 5. Please give the definite range of the eastern TP edge (ETP), northern YGP edge (YGP) and DBM region.]*

**Response 9:** The definite ranges of three regions were defined with the altitudes of 750–3500 m over 30.5–33.0° N, 102.7-105.3 °E (the eastern TP edge), 750–3000 m over 27.8–29 °N, 103.5–108.5 °E (northern YGP edge) and above 750 m over 31.5–33.0 °N, 106.0–109.4 °E (DBM region), which were marked in Figure S5.

The above description has been added in the Sect. 2.3.

[Figure]

**Figure S5**. The eastern TP edge (ETP), northern YGP edge (YGP) and DBM region (black lines) in the study with the terrain heights.

*[9. Line 600: Figure 5(a). Why only 8 hours data are presented here? There is an abnormal value around half past 10 a.m., please give the reason.]*

**Response 10:** The MPL is located at site 15 of the western margin of SCB (Fig. 1). The layer of high PM$_{2.5}$ concentrations with the vertical hollow was observed between 1-2 km during the 8-hr haze maintenance stage P2 at site 15.

The abnormal values as might be caused by the background noise of MPL, giving rise to an aberrant point of observation with extremely high extinction coefficients at this point. The abnormal values around half past 10 a.m. have been removed in the modified Figure 6.

[Figure]

**Figure 6**. Vertical and time cross-sections of $PM_{2.5}$ mass concentrations ($\mu g \ m^{-3}$) from (a) MPL-4B retrievals products and (b) simulation results at site 15 (Fig. 1; Table 1) in the western SCB edge during 7:00 a.m.−2:00 p.m. on 5 January 2017.

*[10. Line 610 -615: Figure 8. Does this cross section along 104.02◦ E? Please specify. Same as previous mentioned, why do you select this site 1 (104.02◦ E; 30.67◦ N)? Do you think it may be a better choice if you put the cross section along the wind vectors from northeast to southwest?]*

**Response 11:** Yes, Figure 9 (in the revised manuscript) actually exhibited the cross sections along the Chengdu (104.02 °E). The vertical cross section along the northeast-southwest wind vectors were provided in Figure S4 (please see our response 4). The separate height-longitude and height-latitude cross sections could better represent the vertical circulation changes and $PM_{2.5}$ distribution over SCB.

*[11. Line 625: Figure 10. How the values of surface PM2.5 concentrations are calculated? The regional average of the SCB or the average of several monitoring sites in SCB?]*

**Response 12:** Both surface $PM_{2.5}$ concentrations and the contribution proportions in Figure 11 (in the revised manuscript) were calculated with the regional averages over SCB rather than the averages of several monitoring sites in cities. The caption of Figure 11 has been revised as follows:

"Figure 11. Hourly variations of surface $PM_{2.5}$ concentrations originated from the SCB's anthropogenic emissions (blue filled areas) and the contribution proportions to the basin surface $PM_{2.5}$ levels (red curve) during 1−8 January 2017 based on the regional averages over SCB. "

---

## Author Response (AR1)

Dear Editors and Referees:

Thank you very much for your review and comments concerning our manuscript entitled "Elevated 3D structures of PM$_{2.5}$ and impact of complex terrain-forcing circulations on heavy haze pollution over Sichuan Basin, China" [MS No.: acp-2020-1161]. Those comments are all valuable and helpful for revising and improving the manuscript. We have studied comments carefully and have accordingly made the revisions. Revised parts are highlighted with Track Changes in the revised manuscript. In the following we quoted each review question in the square brackets and added our response after each paragraph.

==================================================

**Response to Referee #1**

==================================================

*[The paper analyzes an episode with high concentrations of PM2.5 in the Sichuan Basin (China), combining observations and numerical simulations. The paper is potentially interesting, in particular for the peculiar interaction between meso and local circulations and pollutant emissions, which leads to the formation of an elevated pollutant layer. However, the discussion of the results should be improved before the paper can be accepted for publication.]*

**Response 1:** Many thanks for your encouraging comments. We have revised the manuscript accordingly. All the revisions have been highlighted with Track Changes in the revised manuscript. The point-by-point responses to the reviewer's comments are as follows.

*General comments*

*[1. A general meteorological overview of the event, including a synoptic characterization, is missing in the paper.]*

**Response 2:** Following the referee's comment, We have plotted the 700hPa geopotential heights and wind vectors in three stages (newly added Figure 3) over the SCB and the surrounding regions with the meteorology analysis data of ERA-Interim.

We also characterized the overview of the synoptic conditions in the revised Sect. 2.4 as follows:

"The meteorological overview of the haze event was characterized by the 700 hPa fields of geopotential heights and wind vectors (Fig. 3). A trough in the mid-latitude westerlies moved eastward from the eastern edge of the TP to the western SCB margin during P1, the trough of low pressure evolved over the SCB region during P2, and the westerly trough shifted out the SCB region with the low-pressure system disappearing in the P3 (Fig. 3). The changes in atmospheric circulations in the three stages reflected the meteorological modulation of heavy haze development over the SCB in associated with the effect of TP topography on the westerlies. "

[Figure]

**Figure 3**. The 700 hPa geopotential height fields and wind vectors averaged during (a) P1, (b) P2 and (c) P3 stages with the trough line (brown line) and low-pressure center (L). The SCB was outlined with the red solid lines.

From the formation to the maintenance and the dissipation periods of haze pollution, the near-suface prevailing northeasterly winds strengthened gradually over the SCB. During the formation and maintenance stage, the surface winds were weak (1.4–1.7 m s$^{-1}$) over the SCB, which was insufficient to dispel the air pollutants, but to continuously accumulate PM$_{2.5}$ locally from light to heavy pollution conditions (Fig. 7a, Fig. 7b). By the dissipation period, the northeasterly winds intensified and removed PM$_{2.5}$ from the SCB (Fig. 7c).

*[2. The Authors adopt a grid ratio of 1:4, while an odd grid ratio is recommended because for even values interpolation errors arise due to the nature of Arakawa C-grid staggering. The authors should at least discuss this choice.]*

**Response 3:** Thanks for the referee's suggestion. In the revised manuscript, we have accordingly added the following discussions in the revised Sect. 2.3:

"Considering the complex terrain underlying of the SCB's deep basin and surrounding plateaus and mountains in Southwest China, we adopted a grid ratio of 1:4 for simulation experiments with a precisely defined horizontal resolution. It should be noted that the even grid ratio may cause interpolation errors at the nested-domain boundary conditions owing to the nature of Arakawa C-grid staggering.

*[3. The Authors say that the "vertical turbulent diffusion coefficient of the boundary layer was reduced". This aspect should be better discussed, since it might significantly affect the results.]*

**Response 4:** Following the referee's comments, we have added the below discussions into the revised Sect. 2.3.

"High $PM_{2.5}$ levels in the atmosphere could significantly reduce the near-ground solar radiation for stable atmospheric stratification, which decreases the vertical turbulent diffusion in the boundary layer (Wang et al., 2019). This is an important mechanism in the formation of severe haze pollution with the explosive growth of $PM_{2.5}$ (Zhong et al., 2018). The overestimated vertical diffusion capacity under poor air quality conditions (Ren et al., 2019) causes deviations in air pollutant concentrations simulated in air quality models (Wang et al., 2018). In this study, the vertical turbulent diffusion coefficient of the atmospheric boundary layer was cut halfway for better simulation of the 3D structures of $PM_{2.5}$, during the heavy air pollution event over the SCB region."

**References**

Wang, L., Liu, J., Gao, Z., Li, Y., Huang, M., Fan, S., Zhang, X., Yang, Y., Miao, S., Zou, H., Sun, Y., Chen, Y., and Yang, T.: Vertical observations of the atmospheric boundary layer structure over Beijing urban area during air pollution episodes, Atmos. Chem. Phys., 19, 6949–6967, https://doi.org/10.5194/acp-19-6949-2019, 2019.

Zhong, J., Zhang, X., Dong, Y., Wang, Y., Liu, C., Wang, J., Zhang, Y., and Che, H.: Feedback effects of boundary-layer meteorological factors on cumulative explosive growth of $PM_{2.5}$ during winter heavy pollution episodes in Beijing from 2013 to 2016, Atmos. Chem. Phys., 18, 247–258, https://doi.org/10.5194/acp-18-247-2018, 2018.

Ren, Y., Zhang, H., Wei, W., Wu, B., Liu, J., Cai, X., and Song, Y.: Comparison of the turbulence structure during light and heavy haze pollution episodes, Atmospheric Research, 230, 0169-8095, https://doi.org/10.1016/j.atmosres.2019.104645, 2019.

Wang, H., Peng, Y., Zhang, X., Liu, H., Zhang, M., Che, H., Cheng, Y., and Zheng, Y.: Contributions to the explosive growth of $PM_{2.5}$ mass due to aerosol–radiation feedback and decrease in turbulent diffusion during a red alert heavy haze in Beijing–Tianjin–Hebei, China, Atmos. Chem. Phys., 18, 17717–17733, https://doi.org/10.5194/acp-18-17717-2018, 2018.

*[4. No information about the vertical discretization is given. An adequate vertical resolution is fundamental to evaluate the thermal stratification over complex terrain.]*

**Response 5:** Thanks for the referee's comments. The information about the vertical discretization is added in the revised manuscript (lines 114-116) as follows:

"An adequate vertical resolution is fundamental for evaluating thermal stratification over a complex terrain. Therefore, 35 vertical layers were set with fine resolutions of 30–120 m in the boundary layer."

*[5. The Authors propose a series of statistical indexes for evaluating model results, both for meteorological variables and PM2.5. From these statistical indexes it is difficult to judge the performance of the model, regarding in particular the time evolution of observed and simulated variables. I strongly suggest to show some representative time series to better evaluate the model performance at some representative location.]*

**Response 6:** Following the referee's suggestion, the hourly variations of $PM_{2.5}$ concentrations, 2 m air temperature, surface relative humidity and near-surface wind speed in Chengdu (site 1), Suining (site 10) and Zigong (site 13) were shown in Figures S1 and S2 in the supplement of manuscript. The comparisons between

observation and simulation also were evaluated with the reasonable WRF-Chem modeling performance.

[Figure]

**Figure S1**. Hourly variations of observed (black curves) and simulated (red curves) PM$_{2.5}$ concentrations respectively in (a) Chengdu (site 1), (b) Suining (site 10) and (c) Zigong (site 13) during the haze pollution episode.

[Figure]

**Figure S2**. Hourly variations of observed (black curves) and simulated (red curves) 2 m air temperature, surface relative humidity and wind speed results respectively in (a) Chengdu (site 1), (b) Suining (site 10) and (c) Zigong (site 13) during the haze pollution episode in 2017.

*[6. Figure 4 presents a comparison between the vertical profiles of potential temperature, wind speed and relative humidity from observations and model results. Also in this case it is difficult to evaluate model results, since only mean profiles and the variation range over the entire period are presented. I suggest to show also some representative profiles at some specific hours. In particular, it would be interesting to evaluate how the WRF model is able to capture the vertical temperature profile, since atmospheric stability is crucial for pollutant dispersion. In many points in the paper a temperature inversion is cited, but the simulation of this temperature inversion is never discussed. For example, at lines 243-250, "thermo-dynamical structures" and "stable stratification" are cited, but, without a representative figure, it is difficult to follow the discussion of the results.]*

**Response 7:** Following the referee's suggestions, the vertical air temperature profiles were evaluated with the comparisons of vertical air temperature profiles between observation and simulation during different haze periods at (a) 11:00 p.m. on 2 January, (b) 05:00 a.m. on 3 January, (c) 02:00 p.m. on 3 January, (d) 02:00 p.m. on 4 January, (e) 08:00 p.m. on 4 January, (f) 05:00 p.m. on 5 January, (g) 08:00 a.m. on 6 January, (h) 11:00 a.m. on 6 January and (i) 11:00 a.m. on 7 January 2017. (Figure S3 in the supplement of manuscript) with the following description added in the revised Sect. 3.1:

"Compared with the observed air temperature, the WRF-Chem simulations were evaluated to reasonably capture the vertical temperature profiles for understanding atmospheric stability in the vertical thermodynamic structures of the boundary layer over the SCB (Fig. S3)."

We have clarified the description of "thermo-dynamical structures" and "stable stratification" into the revised Sect. 3.3.

"The potential temperature vertical gradients (Fig. 5), which are used for assessing atmospheric stability, were estimated respectively with 4.0 K/km, 7.8 K/km

and 5.2 K/km in the boundary layer during the three periods of haze pollution with near-surface strong temperature inversion (Fig. S3), presenting the thermodynamic structure with stable stratification in the atmospheric boundary layer, weakening the air pollutant dispersion."

[Figure]

**Figure S3**. Comparisons of vertical profiles of air temperature between observation (black curves) and simulation (red curves) at (a) 11:00 p.m. on 2 January, (b) 05:00 a.m. on 3 January, (c) 02:00

p.m. on 3 January, (d) 02:00 p.m. on 4 January, (e) 08:00 p.m. on 4 January, (f) 05:00 p.m. on 5 January, (g) 08:00 a.m. on 6 January, (h) 11:00 a.m. on 6 January and (i) 11:00 a.m. on 7 January 2017.

*[7. Although the paper is rather well written, a review by a native English speaker would be beneficial.]*

**Response 8:** Thanks for your positive comments. A native English speaker in ELSEVIER Language Editing Services has reviewed the paper to improve the language (please see the following certificate).

[Figure]

*Specific comments:*

*[Page 2, line 79: "Section 2 introduced…". Here and in other parts of the paper I would use the present tense (when referring to tables, figures…).]*

**Response 9:** Thanks for the careful review, we have changed with the present tense (when referring to tables, figures…).

*[Figures 7 and 8: the location of the cross sections should be indicated in Fig. 1.]*

Response 10: We have indicated the location of the cross sections in the modified Figure 1.

[Figure]

**Figure 1**. (Left panel) three nesting domains D1, D2 and D3 of WRF-Chem simulation with the terrain heights (m in a.s.l.) and (right panel) the location of 18 urban observation sites (black dots, Table 1) including site 1 (Chengdu) with the intensive sounding observations and site 15 (Ya'an) with the ground-based MPL detection in the SCB with the surrounding Tibetan Plateau (TP), Yunnan-Guizhou Plateau (YGP), Mountains Daba (Mt. Daba) and Wu (Mt. Wu) in Southwest China. The red dash lines indicate the location of the cross sections respectively along 30.67º N and 104.02º E.

====================================================

**Response to Referee #2**

====================================================

*[This paper analyzed the three-dimensional distribution of PM2.5 concentrations in Sichuan Basin during a heavy haze pollution episode in January 2017. The topic is quite interesting; However, many discussions are only general descriptions of phenomena and processes, lacking in-depth analysis and discussion. This makes the article as a whole difficult to follow.]*

**Response 1:** Many thanks for your encouraging comments. We have revised the manuscript accordingly with in-depth analysis and discussion. All the revisions have been highlighted with Track Changes in the revised manuscript. The point-by-point responses to the reviewer's comments are as follows.

*[1. Line 115: The Multi-resolution Emission Inventory for China (MEIC) has been*

*updated to 2017 (http://www.meicmodel.org), while the authors used the data of 2012 for the simulation period in January, 2017. The author should explain the mismatch. Further, the first domain (D1) of the study area includes China and its neighboring countries/regions. In this section, the author only stated that the anthropogenic emission sources used in their study is MEIC data, but as far as I know, MEIC data only includes the anthropogenic emission sources in China, while the emissions from natural sources and neighboring countries/regions are not included. How did the author consider this in their simulation process? If the emission data of neighboring countries/regions are not included, there will be great uncertainty about the results of the section 3.5 (Contribution of local emission and outflow transport), because the surrounding emissions are ignored.]*

**Response 2:** Thanks for the comments. We have accordingly added the following explanation and discussions in the revised conclusions (Sect. 4.):

"MEIC 2017 was not available for the WRF-Chem model. The SCB is located in Southwest China with larger uncertainties in the anthropogenic emission inventory compared to Eastern China. An accurate emission inventory could improve air pollution simulations and air quality change assessments in future studies."

Furthermore, we have clarified the emissions from natural sources and neighboring countries/regions in the revised manuscript (Sect. 3.5) as follows:

"The Model of Emissions of Gases and Aerosols from Nature (v2.1) was applied to the natural emission sources in the simulation with dust emission parameterization. The SCB region in the northeastern part of Southwest China, characterized by a deep-bowl structure, is isolated by plateaus (TP in the west and YGP in the south) and mountains with a clean atmospheric environment. Haze pollution events with extremely high $PM_{2.5}$ concentrations over the SCB are ascribed to the accumulation of local anthropogenic pollutants and air pollutant transport over the basin (Wang et al., 2018; Qiao et al., 2019; Zhao et al., 2019). High local anthropogenic pollutant emissions in the SCB dominate regional air pollution over the SCB (Liao et al., 2017). The transport of air pollutants from neighboring countries in South Asia is mostly concentrated in the neighboring regions of the southern TP and southern YGP (Wang et al., 2018; Zhao et al., 2019; Yin et al., 2020). Therefore, the anthropogenic emission

data of South Asian neighboring countries of China are not included in the WRF-Chem simulation on haze pollution over the SCB during 2–8 January 2017, considering the less effects of northward cross-border transport of air pollutants from South Asian neighboring countries on air pollution in SCB with prevailing northeasterly wind during Asian winter monsoon season with a negligible contribution to the wintertime heavy haze pollution over the SCB region."

**References**

Liao, T., Wang, S., Ai, J., Gui, K., Duan, B., Zhao, Q., Zhang, X., Jiang, W., and Sun, Y.: Heavy pollution episodes, transport pathways and potential sources of $PM_{2.5}$ during the winter of 2013 in Chengdu (China), Science of the Total Environment, 584, 1056-1065, https://doi.org/10.1016/j.scitotenv.2017.01.160, 2017.

Wang, H., Tian, M., Chen, Y., Shi, G., Liu, Y., Yang, F., Zhang, L., Deng, L., Yu, J., Peng, C., and Cao, X.: Seasonal characteristics, formation mechanisms and source origins of $PM_{2.5}$ in two megacities in Sichuan Basin, China, Atmos. Chem. Phys., 18, 865–881, https://doi.org/10.5194/acp-18-865-2018, 2018.

Zhao, S., Yu, Y., Qin, D., Yin, D., Dong, L., and He, J.: Analyses of regional pollution and transportation of $PM_{2.5}$ and ozone in the city clusters of Sichuan Basin, China, Atmospheric Pollution Research, 10(2), 374-385, https://doi.org/10.1016/j.apr.2018.08.014, 2019.

Yin, D., Zhao, S., Qu, J., Yu, Y., Kang, S., Ren, X., Zhang, J., Zou, Y., Dong, L., Li, J., He, J., Li, P., and Qin, D.: The vertical profiles of carbonaceous aerosols and key influencing factors during wintertime over western Sichuan Basin, China, Atmospheric Environment, 223, 1352-2310, https://doi.org/10.1016/j.atmosenv.2020.117269, 2020.

*[2. Line 157: 3.1 Model evaluation As we know, China has adopted active pollution source control policies in the last 5 years, and the intensity, the temporal and spatial distribution of emission sources will vary greatly from year to year. The author selected the 2012 MEIC inventory as its emission data. Thus the model evaluation result may not be convincing.]*

**Response 3:** We agree with the referee that China has adopted active pollution source control policies in the last 5 years, and the intensity, the temporal and spatial distribution of emission sources will vary greatly from year to year. We have added the explanation and discussions about the 2012 MEIC emission data in the revised conclusions as follows:

"MEIC 2017 was not available for the WRF-Chem model. The SCB is located in Southwest China, with larger uncertainties in the anthropogenic emission inventory compared to Eastern China, An accurate emission inventory could improve air pollution simulations and air quality change assessments in future studies."

*[3. Line 216: To examine the vertical structures of PM2.5 concentrations over SCB, we selected the urban site 1 (104.02° E; 30.67° N) in Chengdu (cf. Fig. 1) as a reference point to investigate the distributions of PM2.5 and the atmospheric circulations respectively in the vertical-meridional and vertical-zonal cross-sections. Why do you select the urban site 1 (104.02° E; 30.67° N) in Chengdu for the vertical discussion. Do you have any special purpose? Chengdu is located in the far west side of the SCB, and other sites in the central area of SCB maybe are better choices, as the wind vectors shows in Figure 6.]*

**Response 4:** We selected the urban site 1 (104.02° E; 30.67° N) in Chengdu for the vertical discussion with the following purposes:

1)    The terrain effect of TP, the "world roof" on the mid-latitude westerlies could modulate haze pollution in the downstream region over China (Xu et al., 2016). The SCB is immediately to the east of TP with a large elevation drop exceeding 3000 m over a short horizontal distance. The unique terrain effect generates the asymmetries of meteorological and air pollutant distribution over the SCB (Zhang et al., 2019). Chengdu on the far west side of the SCB was selected to better understand the elevated 3D structures of $PM_{2.5}$ with the impact of TP terrain-forcing circulations on the haze pollution episode over the SCB.

2)    Chengdu (site 1), is a metropolis in the SCB with high anthropogenic emissions and the most polluted environment in Southwest China (Ning et al., 2018). It is important to investigate how the urban surface high $PM_{2.5}$ levels evolved vertically in the atmosphere with the combination between the high urban emissions and TP's terrain-forcing lifting over the SCB.

Furthermore, we have plotted the cross-sections of $PM_{2.5}$ and wind vectors along the near-surface prevailing northeastern wind across the central SCB (blue line in

Figure S4 of manuscript supplement). The vertical changes of PM$_{2.5}$ with the terrain-forcing local circulations by YGP-terrain effects were remarkably presented in the different stages of the heavy haze pollution event.

[Figure]

**Figure S4**. Northeast-southwest cross-sections along the near-surface prevailing wind (blue line) over the SCB (left panel), PM$_{2.5}$ concentrations (color contours: μg m$^{-3}$) and wind vectors (right panel) in (a) the relative clean environment at 12:00 a.m. on 2 January, (b) heavy air pollution formation stage at 12:00 a.m. on 3 January, and (c) maintenance stage at 8:00 a.m.on 6 January 6 and (d) dissipation stage at 8:00 a.m. on 7 January, 2017.

**References**

Xu, X., Zhao, T., Liu, F., Gong, S. L., Kristovich, D., Lu, C., Guo, Y., Cheng, X., Wang, Y., and Ding, G.: Climate modulation of the Tibetan Plateau on haze in China, Atmos. Chem. Phys., 16, 1365–1375, https://doi.org/10.5194/acp-16-1365-2016, 2016.

Zhang, L., Guo, X., Zhao, T., Gong, S., Xu, X., Li, Y., Luo, L., Gui, K., Wang, H., Zheng, Y, and Yin, X.: A modelling study of the terrain effects on haze pollution in the Sichuan Basin, Atmos. Environ., 196, 515 77–85, https://doi.org/10.1016/j.atmosenv.2018.10.007, 2019.

Ning, G., Wang, S., Ma, M., Ni, C., Shang, Z., Wang, J. and Li, J.: Characteristics of air pollution in different zones of Sichuan Basin, China, Sci. Total. Environ., 612, 975–984, 440 https://doi.org/10.1016/j.scitotenv.2017.08.205, 2018.

*[4. Line 275: Figure 10 showed the PM2.5 concentrations emitted from the regional air pollutant sources over the SCB region and the relative contribution rates to air pollution changes.*
*The expression here shows the author's conceptual misunderstanding of the source of PM2.5. How can the "PM2.5 concentrations" be "emitted"?]*

**Response 5:** We modified the expression as "Figure 11 shows the $PM_{2.5}$ concentrations originating from local emissions of primary $PM_{2.5}$, gaseous precursors of $PM_{2.5}$ over SCB and the relative contribution rates to air pollution changes."

*[5. Line 278: The SCB's regional air pollutant emissions provided surface PM2.5 concentrations from 40.6 to 136.2 µg m-3, contributing 75.4–94.6 % of surface PM2.5 concentrations for the heavy pollution episode over SCB, indicating its dominant role over this isolated deep basin in Southwest China.*
*What does "indicating its dominant role over this isolated deep basin in Southwest China" mean? It is hard to follow.]*

**Response 6:** In the revised manuscript, we have clarified this sentence as "The SCB's regional air pollutant emissions provided surface $PM_{2.5}$ from 40.6 to 136.2 µg m-3, contributing 75.4–94.6% of total concentrations for the heavy pollution episode over SCB. This indicates the dominant role of local air pollutant emissions on air quality changes over this isolated deep basin in Southwest China."

*[6. Line 279: The regionally emitting PM2.5 concentrations averaged over SCB were 88.64, 91.04 and 65.96 µg m-3 for the formation, maintenance and dissipation periods, respectively.*
*Same as above. How can the "PM2.5 concentrations" be "emitted"?]*

**Response 7:** The expression has been corrected as "The surface $PM_{2.5}$ concentrations sourced from the regional air pollutant emissions over the SCB were averaged, with 88.64, 91.04 and 65.96 µg m$^{-3}$ for P1, P2, and P3, respectively."

*[7. Line 284: We think the exchanges of PM2.5 between the polluted air over SCB and the cleaner environment air over the surrounding plateaus and mountains in Southwest China play a role in this process. (Figs. 7 and 8).*
*How do you think the PM2.5 can be "exchanged" between the polluted air over SCB and the cleaner environment air?]*

**Response 8:** Thanks for the referee's careful review. The sentence has been modified as "This could be attributed to the exchanges between the $PM_{2.5}$-rich airmass over SCB and $PM_{2.5}$-poor airmass in the surrounding plateaus and mountains over

Southwest China (Fig. 8 and 9)."

*[8. Line 560: Table 5. Please give the definite range of the eastern TP edge (ETP), northern YGP edge (YGP) and DBM region.]*

**Response 9:** The definite ranges of the three regions were defined with the altitudes of 750–3500 m over 30.5–33.0° N, 102.7-105.3 °E (the eastern TP edge), 750–3000 m over 27.8–29 °N, 103.5–108.5 °E (northern YGP edge) and above 750 m over 31.5–33.0 °N, 106.0–109.4 °E (DBM region), as shown in Fig. S5.

The above description has been added in the Sect. 2.3.

[Figure]

**Figure S5**. The roughly periphery of eastern TP edge (ETP), northern YGP edge (YGP) and DBM region (black lines) in the study with the terrain heights.

*[9. Line 600: Figure 5(a). Why only 8 hours data are presented here? There is an abnormal value around half past 10 a.m., please give the reason.]*

**Response 10:** The MPL is located at site 15 of the western margin of SCB (Fig. 1). The layer of high $PM_{2.5}$ concentrations with the vertical hollow was observed between 1–2 km during the 8-hr haze maintenance stage P2 at site 15.

The abnormal values as might be caused by the background noise of MPL,

giving rise to an aberrant point of observation with extremely high extinction coefficients at this point. The abnormal values around half past 10 a.m. have been removed in the modified Figure 6.

[Figure]

**Figure 6**. Vertical and time cross-sections of PM$_{2.5}$ mass concentrations ($\mu$g m$^{-3}$) from (a)

MPL-4B retrievals products and (b) simulation results at site 15 (Fig. 1; Table 1) in the western SCB edge during 7:00 a.m.–2:00 p.m. on 5 January 2017.

*[10. Line 610 -615: Figure 8. Does this cross section along 104.02◦ E? Please specify. Same as previous mentioned, why do you select this site 1 (104.02◦ E; 30.67◦ N)? Do you think it may be a better choice if you put the cross section along the wind vectors from northeast to southwest?]*

**Response 11:** Yes, Figure 9 (in the revised manuscript) actually exhibited the cross sections along the Chengdu (104.02 °E). The vertical cross section along the northeast-southwest wind vectors were provided in Figure S4 (please see our response 4). The separate height-longitude and height-latitude cross sections could better represent the vertical circulation changes and PM$_{2.5}$ distribution over the SCB.

*[11. Line 625: Figure 10. How the values of surface PM2.5 concentrations are calculated? The regional average of the SCB or the average of several monitoring sites in SCB?]*

**Response 12:** Both surface PM$_{2.5}$ concentrations and the contribution proportions in Figure 11 (in the revised manuscript) were calculated with the regional averages over the SCB rather than the averages of several monitoring sites in cities. The caption of Figure 11 has been revised as follows:

"Figure 11. Hourly variations of surface PM$_{2.5}$ concentrations originating from the SCB's anthropogenic emissions (blue filled areas) and the contribution proportions to the basin surface PM$_{2.5}$ levels (red curve) during 1–8 January 2017 based on the regional averages over the SCB. "

---

## Author Response (AR2)

Dear Editors and Referees:

Thank you very much for your review and comments concerning our manuscript entitled "Elevated 3D structures of $PM_{2.5}$ and impact of complex terrain-forcing circulations on heavy haze pollution over Sichuan Basin, China" [MS No.: acp-2020-1161]. We have revised the manuscript accordingly. All the revisions have been highlighted with Track Changes in the revised manuscript. In the following, we quoted each review question in the square brackets and added our response after each paragraph.

================================================

**Response to Editor:**

================================================

*[1.Concerning the model configuration: It would be interesting to know how many vertical levels you used in the model above the boundary level, and what is your model top. Please add those information in the manuscript.]*

**Response 1:** Thanks for your comments. we have revised the sentence in Sect. 2.3. "Therefore, 35 vertical layers from the ground to the model top at 100 hPa were set for the modeling experiments in this study on air pollution change with the 18 layers in the fine resolutions of 30–120 m vertically from the ground to 1km within the atmospheric boundary layer."

*[2. Furthermore, you don't explain why you use MYJ as a boundary layer scheme. There are other schemes in WRF, such as MYNN, which might have a different turbulent diffusion coefficient and a different treatment of the PBL dynamics. Please add a reference that have used MYJ for a similar work in a complex terrain environment (e.g. in California or Colorado in the USA or over China).]*

**Response 2:** Thank the referee for the suggestion. There are various planetary boundary layer schemes in WRF, including the nonlocal closure schemes (MRF, YSU and SH schemes) and local closure schemes (e.g., MY series schemes). Aims at the stable atmospheric stratification and weak turbulent mixing over the complex terrain,

such as California (Lu et al., 2012), Jharkhand state of India (Madala et al., 2015) and Beijing-Tianjin-Hebei region in China (Bei et al., 2019), the MYJ produced better model performance. Following your suggestion, the explanation on the MYJ scheme has been revised in Sect. 2.3 as follows:

"The MYJ is a local closure scheme (Janjić, 1994), which is applicable to the atmospheric environment with stable atmospheric stratification for weak turbulent mixing (Jia and Zhang, 2020) and underlying complex terrain (Lu et al., 2012; Madala et al., 2015; Bei et al., 2019). Therefore, the MYJ was used as the planetary boundary layer parameterization scheme in the simulation."

**Table 2**. Setting of physical and chemistry schemes in the WRF-Chem simulations

| | |
|---|---|
| Microphysics | Morrison 2-mom |
| Boundary layer | MYJ |
| Longwave radiation | RRTM |
| Shortwave radiation | RRTMG |
| Land surface | Noah |
| Cumulus convection | Grell 3D (none in D3) |
| Urban scheme | Single-layer |
| Chemistry | RADM2 |
| Aerosol particles | MADE/SORGAM |
| Photolysis | Madronich (TUV) |

*[3. Figure S1 and S2 are not discussed in the paper. Please add a comment on those figures in the main article.]*

**Response 3:** Thanks for your comments. In the revised manuscript, we have accordingly modified in Sect. 3.1:

"First, we validated the WRF-Chem simulation performance by comparing with the meteorological and $PM_{2.5}$ observations in the SCB, especially with the intensive vertical soundings, for verifying the vertical structures of the simulated boundary layer (Fig. S1-S3)."

*[4. Figure S3: please add the date and time information on top of each subplots.]*

**Response 4:** Thanks for your suggestion, we have added the date and time information on each subplot in Figure S3 as follows:

[Figure]

**Figure S3**. Comparisons of vertical profiles of air temperature between observation (black curves) and simulation (red curves).

*[5. Table 3: add the units to T2, RH and WS10, and define in the caption what it refers to]*

**Response 5:** We have modified the caption of Table 3 and added the responding units.

**Table 3**. Statistical metrics of comparisons between simulated (Sim.) and observed (Obs.) 2-m air temperature ( T2), surface relative humidity (RH) and 10-m wind speed (WS10) with the correlation coefficient (R), mean bias(MB), mean error (ME) and root mean squared error (RMSE) during air pollution process over 2–7 January 2017.

| | Obs. | Sim. | R | MB | ME | RMSE |
|---|---|---|---|---|---|---|
| T2 | 9.9℃ | 9.2℃ | 0.78** | −0.7 | 1.7 | 2.1 |
| RH | 85.1 % | 77.7 % | 0.67** | −7.4 | 11.2 | 13.4 |
| WS10 | 1.2 m s⁻¹ | 1.5 m s⁻¹ | 0.41* | 0.3 | 0.8 | 1.1 |

Note: MB, ME, RMSE were calculated as following: $MB = \frac{1}{N}\sum_{i=1}^{N}(M_i - O_i)$; $ME = \frac{1}{N}\sum_{i=1}^{N}|M_i - O_i|$; $RMSE = \sqrt{\frac{1}{N}\sum_{i=1}^{N}(M_i - O_i)^2}$; (M and O represented the results from simulation and observation). The ** and * respectively indicated the correlation coefficients R passing the 99% and 95% significant test.

=======================================================

**Response to Referee**

=======================================================

*[1. Meteorological conditions*

*Given the synoptic conditions, I still do not understand the reason for the increase of northeasterly winds during the dissipation stage. A more detailed description of these mechanisms would be in my opinion beneficial for the reader not expert of the meteorology of this region.]*

**Response 1:** The description of synoptic mechanisms was added in the revised Sect. 2.4 as follows:

"Under the typical Asian monsoon climate in January over the SCB, the change of synoptic conditions during the haze event over the SCB were characterized by the cold air invasion driven by near-surface northeasterly winds with the vertical

configuration of trough development and movement in the mid-latitude westerlies at 700 hPa (Figs. 3 and 7). A 700 hPa trough in the mid-latitude westerlies moved eastward from the eastern edge of the TP to the western SCB margin during P1 (Fig. 3a), the trough of low pressure evolved at 700 hPa during P2 (Fig. 3b), and the 700-hPa trough and the low-pressure system disappeared in P3 over the SCB (Fig. 3c). Meteorologically, the direction and intensity of cold air invasion with near-surface northeasterly winds are steered by the development and movement of the westerly trough in the mid-troposphere (Fig. 3c). The 700-hPa trough approached, developed and disappeared in P1, P2 and P3 of the haze pollution event over the SCB (Fig. 3), which is associated with the increase of northeasterly winds for the cold air invasion to the SCB region during the dissipation stage. The changes in atmospheric circulations in the three stages reflected the meteorological modulation of heavy haze development over the SCB in association with the effect of TP topography on the westerlies."

*[2. Model set-up*

*Some choices in the model set-up are still not clear. The explanation of the choice of the 1:4 grid ratio is not clear. Why the presence of complex terrain should be the explanation? Moreover, it is not clear on which basis the Authors decided to cut half the turbulent diffusion coefficient. Did You run some sensitivity tests?]*

**Response 2:** The description of the model set-up and the choice of the 1:4 grid ratio have been modified as follows, and the revised Table 2 is also listed. We did the sensitivity tests with changing turbulent diffusion coefficient, and the test with cutting half the turbulent diffusion coefficient was validated with the most reasonable simulation of air pollutants over the SCB.

[revised manuscript text omitted]

*[3. Results*

*Section 3.3 should be re-organized since it is not clear and many concepts are repeated at different points with different words.Moreover, from Fig. 8 I cannot see the differences between the different phases highlighted by the Authors at lines 269-275 (PM2.5 elevated to the free atmosphere in clean environment and dissipation periods and pressed down in formation and maintenance periods). For example, I can see upward arrows in Fig. 8c, where the Authors indicate the downward branch of the vortex. The most significant difference that I can see in Figure 8 is the stronger low-level wind in Fig. 8d (but I cannot understand the cause of this wind, see above)]*

**Response 3:** Thanks for the referee's suggestions. According to the suggestions, 1) we have revised the original lines 269-275 with "Similarly, $PM_{2.5}$ elevated to the free atmosphere in a clean environment and dissipation periods and pressed down in formation and maintenance periods", 2) we have modified the upward and downward arrows in the revised Fig. 8c, and 3) added the cause of the stronger low-level winds in Fig. 8d (please also see the response 1).

[Figure]

**Figure 8**. Height-longitude cross-sections of PM$_{2.5}$ concentrations (color contours: μg m$^{-3}$) and wind vectors along 30.67º N in the (a) clean environment at 12:00 a.m. on 2 January 2017 (b) heavy air pollution formation stage at 12:00 a.m. on 3 January 2017 (c) maintenance stage at 8:00 a.m.on 6 January 2017, and (d) dissipation stage at 8:00 a.m. on 7 January 2017. The brown arrows highlighted the major air flows (red arrows) associated with the terrain of TP, SCB and Mt. Wu (filled brown color).

In the revised manuscript, section 3.3 has been accordingly re-organized and modified with the above discussions as follows:

[revised manuscript text omitted]

*[4. Page 6, line 160: the date of the end of the dissipation stage is wrong.]*

**Response 4:** Thanks for your careful review. The concentrations of $PM_{2.5}$ pollution event indeed ended on 7 January 2017 and there is an error in our description here. We have corrected it accordingly in the manuscript.

**Response 5:** The caption of Figure 9 has been modified as:

"Same as Fig. 8, but for height-latitude cross-sections of $PM_{2.5}$ concentrations and wind vectors."